# Dissecting Mechanisms of Melanoma Resistance to BRAF and MEK Inhibitors Revealed Genetic and Non-Genetic Patient- and Drug-Specific Alterations and Remarkable Phenotypic Plasticity

**DOI:** 10.3390/cells9010142

**Published:** 2020-01-07

**Authors:** Mariusz L. Hartman, Malgorzata Sztiller-Sikorska, Anna Gajos-Michniewicz, Malgorzata Czyz

**Affiliations:** Department of Molecular Biology of Cancer, Medical University of Lodz, 6/8 Mazowiecka Street, 92-215 Lodz, Poland; mariusz.hartman@umed.lodz.pl (M.L.H.); malgorzata.sztiller-sikorska@umed.lodz.pl (M.S.-S.); anna.gajos-michniewicz@umed.lodz.pl (A.G.-M.)

**Keywords:** acquired resistance, vemurafenib, trametinib, melanoma plasticity, reversible transcriptional reprogramming, growth factor dependence, AXL, NGFR, RBMX, patient-to-patient variability

## Abstract

The clinical benefit of MAPK pathway inhibition in BRAF-mutant melanoma patients is limited by the development of acquired resistance. Using drug-naïve cell lines derived from tumor specimens, we established a preclinical model of melanoma resistance to vemurafenib or trametinib to provide insight into resistance mechanisms. Dissecting the mechanisms accompanying the development of resistance, we have shown that (i) most of genetic and non-genetic alterations are triggered in a cell line- and/or drug-specific manner; (ii) several changes previously assigned to the development of resistance are induced as the immediate response to the extent measurable at the bulk levels; (iii) reprogramming observed in cross-resistance experiments and growth factor-dependence restricted by the drug presence indicate that phenotypic plasticity of melanoma cells largely contributes to the sustained resistance. Whole-exome sequencing revealed novel genetic alterations, including a frameshift variant of RBMX found exclusively in phospho-AKT^high^ resistant cell lines. There was no similar pattern of phenotypic alterations among eleven resistant cell lines, including expression/activity of crucial regulators, such as MITF, AXL, SOX, and NGFR, which suggests that patient-to-patient variability is richer and more nuanced than previously described. This diversity should be considered during the development of new strategies to circumvent the acquired resistance to targeted therapies.

## 1. Introduction

A constitutive activation of the mitogen-activated protein kinase (MAPK) signaling pathway due to genetic alterations is detected in the majority of melanomas [1]. Addiction of melanoma cells to the MAPK cascade-related oncogene activity is therapeutically targeted by selective inhibitors of mutated B-RAF proto-oncogene (BRAFmut) and mitogen-activated protein kinase 1/2 (MEK1/2) that have produced a robust response of melanoma patients [2,3,4]. Durability of the clinical benefit is a major obstacle because of acquired resistance [5,6,7]. Although acquired resistance may be driven by genetic alterations exemplified by mutations of genes encoding neuroblastoma Ras viral oncogene homolog (NRAS), Kirsten rat sarcoma viral proto-oncogene (KRAS), MEK1/2 and extracellular signal-regulated kinase 1/2 (ERK1/2), amplification of *BRAF* and alternative splicing of BRAF transcript, loss of cyclin-dependent kinase inhibitor 2A (CDKN2A) and alterations of genes encoding the components of the phosphoinositide-3-kinase–protein kinase B/AKT (PI3K/AKT) signaling pathway [8,9,10,11,12,13,14], almost 40% of relapsed melanomas do not harbor defined mutational background of resistance despite considerable transcriptomic alterations [15]. Cell plasticity observed as phenotypic transitions towards diverse cell states via epigenetic and transcriptional reprogramming is remarkable in melanomas and largely contributes to drug resistance [16]. Diverse strategies to associate genomic and transcriptomic data with clinical characteristics of patients undergoing treatment and to find an ambiguous biomarker(s) useful for identification of patients who will benefit from durable response to treatment are under consideration [17,18,19,20].

To address the challenges of extensive variability of acquired resistance mechanisms, we have taken the advantage of melanoma cell lines derived from tumor specimens to obtain their counterparts resistant to either vemurafenib (PLX; an inhibitor of BRAF^V600E^) or trametinib (TRA; an inhibitor of MEK1/2). The original drug-naïve melanoma cell lines were previously characterized at the level of cell morphology, proliferation rate, activity of multiple signaling pathways and response to changes in the microenvironment [21,22,23,24,25,26], extended by exome sequencing that has recently indicated a number of genetic variants underlying diverse cell phenotypes [23]. In addition, we have recently shown that the acquisition of resistance to vemurafenib or trametinib is related to either reversible or irreversible dedifferentiation associated with changes in the level of microphthalmia-associated transcription factor (MITF) [27]. In the present study, we provide a comprehensive characteristics of drug-resistant melanoma cell lines by an integration of whole-exome sequencing with molecular and cellular assessment of acquired phenotypes. Moreover, we compared phenotypic changes induced in the initial phase of response to vemurafenib or trametinib and genetic/phenotypic alterations in the acquired drug-resistant phase, in which melanoma cells are capable to proliferate in the presence of drugs at high concentrations.

## 2. Materials and Methods

### 2.1. Cultures of Drug-Naïve and Drug-Resistant Cell Lines

Patient-derived drug-naïve cells were used to obtain drug-resistant melanoma cell lines [28]. The study was approved by Ethical Commission of Medical University of Lodz (identification code: RNN/84/09/KE). Each patient signed an informed consent before tissue acquisition. Drug-naïve cell lines were named after Department of Molecular Biology of Cancer (DMBC) with consecutive numbers. They were grown in vitro in stem cell medium (SCM) consisting of Dulbecco’s Modified Eagle’s Medium (DMEM)/F12 (Lonza, Basel, Switzerland), B-27 supplement (Gibco, Paisley, UK), 10 ng/mL basic fibroblast growth factor (bFGF) (Corning, Corning, NY, USA), 20 ng/mL epidermal growth factor (EGF) (Corning, Corning, NY, USA), insulin (10 µg/mL) (Sigma-Aldrich, St Louis, MO, USA), heparin (1 ng/mL) (Sigma-Aldrich, St Louis, MO, USA), 100 IU/mL penicillin, 100 µg/mL streptomycin, and 2 µg/mL fungizone B. To obtain cell lines resistant to vemurafenib (PLX) or trametinib (TRA) (Selleck Chemicals LLC, Houston, TX, USA), melanoma cell lines derived from different tumor specimens were cultured in SCM in the presence of increasing concentrations of respective drug for 4–5 months. 

### 2.2. DNA Extraction, Whole-Exome Sequencing (WES) and WES Data Analysis

Whole-exome sequencing and data analysis were performed as described previously [23,27]. Target coverage exceeded 100x for all DNA samples (Appendix A). Sequencing data for drug-naïve cell lines can be found under the numbers E-MTAB-6978 at ArrayExpress and ERP109743 at European Nucleotide Archive (ENA). Sequencing data for resistant cell lines are available under the accession numbers: E-MTAB-7248 at ArrayExpress and ERP111109 at European Nucleotide Archive (21_PLXR, 21_TRAR, 28_PLXR, 28_TRAR, 29_PLXR, 29_TRAR and 17_TRAR), E-MTAB-7991 at ArrayExpress and ERP115432 at European Nucleotide Archive (11_PLXR, 11_TRAR, 12_PLXR), and E-MTAB-8150 at ArrayExpress and ERP116314 at European Nucleotide Archive (33_PLXR). VCF files were generated to identify somatic single nucleotide polymorphisms (SNPs) and short insertions or deletions (InDels). Functional effects of SNPs were predicted in silico by the Polyphen-2 software freely available online (genetics.bwh.harvard.edu/pph2/index.shtml). The Polyphen-2-based predictions were classified as benign (scores 0.000–0.449), possibly damaging (scores 0.450–0.959) or probably damaging (scores 0.960–1.000).

### 2.3. Acid Phosphatase Activity (APA) Assay

To assess the viable cell number, the activity of acid phosphatase was measured colorimetrically as described previously [28]. Briefly, melanoma cells were grown for 0, 24, 48, and 72 h, then the plates were centrifuged and medium was replaced with 100 µL of assay buffer containing 0.1 mol/L sodium acetate (pH = 5), 0.1% Triton X-100, and 5 mmol/L p-nitrophenyl phosphate (Sigma-Aldrich, St Louis, MO, USA). The plates were incubated for 2 h at 37 °C. The reaction was stopped by adding 10 µL of 1 mol/L NaOH. The absorbance values were measured using a microplate reader Infinite M200Pro (Tecan, Salzburg, Austria). To study growth factor dependence, melanoma cells were cultured in the medium consisting of DMEM/F12, B-27, insulin, heparin and antibiotics supplemented with 10 ng/mL bFGF, 20 ng/mL EGF and 40 ng/mL HGF (Gibco, Frederick, MD, USA), used either alone or in combination, or they were deprived of exogenous growth factors (noGF medium).

### 2.4. Cell Lysates and Western Blotting

Preparation of cell lysates, electrophoresis, protein transfer, and immunodetection were performed according to the procedures described previously [23,25]. Briefly, cells were lysed for 30 min at 4 °C in RIPA buffer, and centrifuged. Protein concentration was determined by Bradford assay (Biorad, Hercules, CA, USA). Cell lysates were diluted in 2× Laemmli buffer and 15 μg of proteins were loaded on a standard 7% SDS-polyacrylamide gel. The proteins were transferred onto Immobilon-P PVDF membrane (Millipore, Billerica, MA, USA) using BioRad transfer equipment, and transfer efficiency was confirmed by Ponceau S staining. The membrane was incubated in blocking solution for 1 h. Primary antibodies detecting phospho-ERK1/2 (Thr^202^/Tyr^204^), ERK1/2, phospho-MEK1/2 (Ser^217^/Ser^221^), p-AKT (Ser^473^), AKT, AXL, SOX2, EGFR (Cell Signaling, Danvers, MA, USA), BRAF, DUSP6, PARP, SOX10, GAPDH (Santa Cruz Biotechnology, Santa Cruz, CA, USA), active β-catenin (dephosphorylated on Ser^37^ and Thr^41^) from Millipore or β-actin (Sigma-Aldrich, St Louis, MO, USA) were used followed by binding of the secondary HRP-conjugated anti-mouse or anti-rabbit antibodies (Santa Cruz Biotechnology, Santa Cruz, CA, USA). The membranes were incubated with Pierce^®^ ECL Western Blotting Substrate (Pierce, Rockford, IL, USA), and the proteins were visualized by using ChemiDoc Imaging System (Biorad, Hercules, CA, USA), or on a medical X-ray film (Foton-Bis, Bydgoszcz, Poland).

### 2.5. RNA Isolation, cDNA Synthesis, and Quantitative PCR

Total RNA isolation, cDNA synthesis, quantitative real-time polymerase chain reaction (Real-Time PCR) procedures and data analysis were described elsewhere [29]. In brief, total RNA was extracted and purified using Total RNA Isolation kit (A & A Biotechnology, Gdynia, Poland). 1 μg of total RNA was transcribed into cDNA using 300 ng of random primers and SuperScript II Reverse Transcriptase (Invitrogen Thermo Fisher Scientific, Carlsbad, CA,USA). The evaluation of mRNA expression of selected genes was performed by quantitative PCR using the Rotor-Gene 3000 Real-Time DNA analysis system (Corbett Research, Morklake, Australia). Amplification was run using KAPA SYBR FAST qPCR Kit Universal 2x qPCR Master Mix (Kapa Biosystems, Cape Town, South Africa), 200 nM of each primer and 25 ng cDNA template per reaction. The annealing temperature for all genes was 56 °C. The primer sequences for CCND1 and BRAF were published previously [22,30]. Other primers were as following: *AXL*: 5**’**-GGTGGCTGTGAAGATGA-3′ and 5′-CTC AGA TAC TCC ATG CCA CT-3′; *IL-8*: 5′-AGG TGC AGT TTT GCC AAG GA-3′ and 5′-TTT CTG TGT TGG CGC AGT GT-3′; *MMP-2*: 5′-AGC GAG TGG ATG CCG CCT TTA A-3′ and 5′-CAT TCC AGG CAT CTG CGA TGA G-3′; *SOX2*: 5′-GCT AGT CTC CAA GCG ACG AA-3′ and 5′-GCA AGA AGC CTC TCC TTG AA-3′; *SOX9*: 5′-GAC CCG CAC TTG CAC AAC-3′ and 5′-TCC GCT CTC GTT CAG AAG TC-3′. Relative gene expression was calculated versus a reference gene *RPS17*, and efficiency correction for real-time PCR was included.

### 2.6. Human Phospho-Kinase Array

Analysis of 43 phospho-proteins by using Human Phospho-Kinase Array (R & D Systems, Minneapolis, MN, USA) was performed according to the manufacturer’s protocol. In brief, drug-resistant cell lines and matched drug-naïve cell lines cultured in the same conditions for 19 h were used to prepare cell lysates. Antibody-coated membranes were incubated with Array Buffer followed by overnight incubation with cell lysates. After three washes, the membranes were incubated with biotinylated secondary antibodies and streptavidin conjugated with HRP. Chemiluminescence signal was developed on X-ray film or by using ChemiDoc Imaging System (Biorad, Hercules, CA, USA). Pixel densities of dots were analyzed using a transmission-mode scanner and ImageJ free software. Mean pixel densities of the pair of duplicate spots representing each protein of interest and HSP60 protein (a loading control) were calculated. Relative level of each protein was calculated by using the formula: (mean pixel density of the protein)/(mean pixel density of HSP60). Obtained values were used to prepare color coded matrix.

### 2.7. Flow Cytometry

LIVE/DEAD Fixable Violet Dead Cell Stain Kit (Life Technologies, Eugene, OR, USA) was used to exclude dead cells from the analysis. Antibodies against NGFR (PE-conjugated) were from BD Pharmingen (San Jose, CA, USA). Appropriate isotype controls were included in each experiment. Flow cytometric data were acquired with FACSVerse (BD Biosciences, San Jose, CA, USA) and analyzed using BD FACSuite.

### 2.8. Enzyme-Linked Immunosorbent Assay (ELISA)

ELISA kit Quantikine High Sensitivity Human CXCL8/IL-8 (R & D Systems, Minneapolis, MN, USA) was used to determine IL-8 secretion to the culture medium by melanoma cells. Briefly, culture media were diluted 40x, added to each well of the pre-coated plate and incubated for 2 h at room temperature. After 6 washes, the plate was incubated with IL-8 HS Conjugate for 1 h, and washing steps were repeated. This was followed by an incubation with the Substrate Solution, Amplifier Solution and Stop Solution. The optical density of each well was determined using a microplate reader Infinite M200Pro (Tecan, Salzburg, Austria). Concentrations of IL-8 in the medium samples were obtained by applying a four parameter logistic (4-PL) curve fit, and normalized to cell number.

### 2.9. Analysis of AXL Expression Reported in Data Sets from the GEO Database

The publicly available microarray data sets (accession numbers: GSE50509, GSE99898, GSE61992 and GSE77940) were downloaded from the GEO database (http://www.ncb.nlm.nih.gov), and gene expression values were log2-transformed. The AXL expression profiles were developed from paired tumor specimens collected from patients before treatment and post-relapse with acquired resistance either to vemurafenib or combined treatment, dabrafenib and trametinib.

### 2.10. Statistical Analysis

Graphs are presented as mean ± SD and they are representative of 3 biological replicates unless stated otherwise. Student’s *t*-test was performed to determine significant differences between tested parameters. The difference was considered significant if *p* < 0.05.

## 3. Results

### 3.1. The Ability of Developing Resistance to Vemurafenib (PLX) or Trametinib (TRA) Differs between Cell Lines Derived from Different Melanoma Specimens

BRAF^V600E^ melanoma cell lines (DMBC11, DMBC12, DMBC21, DMBC28, DMBC29 and DMBC33) derived from six different tumor specimens were used to develop resistance to vemurafenib (PLX), whereas to obtain melanoma cell lines resistant to trametinib (TRA), in addition to BRAF^V600E^ melanoma cell lines, one Harvey rat sarcoma viral oncogene homolog (HRAS^Q61R^) cell line (DMBC17) was employed. After 4–5 months of continuous exposure to increasing concentrations of PLX and TRA finally reaching 10 µM and 50 nM, respectively, six PLX- (11_PLXR, 12_PLXR, 21_PLXR, 28_PLXR, 29_PLXR, and 33_PLXR) and five TRA-resistant cell lines (11_TRAR, 21_TRAR, 28_TRAR, 29_TRAR, and 17_TRAR) were obtained. None of two attempts to obtain 12_TRAR and 33_TRAR was successful. 33_PLXR cell line derived from DMBC33, which originally exerted very high doubling time (107.7 h) [22], although viable for several weeks did not proliferate efficiently enough to perform all experiments, and 12_PLXR survived only for a few weeks.

### 3.2. Drug-Naïve and Drug-Resistant Cells Differ in Transcript Levels of Cyclin D1, MMP-2, and IL-8

Increasing numbers of viable cells over time indicate that PLXR and TRAR cells became insensitive to drugs in contrast to their original counterparts (Figure 1A). For the pairs DMBC29 vs. 29_TRAR and DMBC17 vs. 17_TRAR, the difference in proliferation curves between drug-naïve and drug-resistant cell lines was less pronounced than in other paired melanoma cell lines. This is probably because these two drug-naïve cell lines and their drug-resistant counterparts exerted high frequencies of cells expressing MITF and several differentiation/pigmentation-related markers at high levels [27]. While we have previously reported that cytostatic effect of vemurafenib and trametinib on drug-naïve cells was uniformly accompanied by a significant decrease in the level of cyclin D1 mRNA [22], the capacity of resistant cells to proliferate in the presence of drug could not be associated with the level of *CCND1* expression. Proliferation rates of resistant cells were high even if all PLXR cell lines had significantly lower transcript levels of cyclin D1 than their drug-naïve counterparts, whereas in TRAR cell lines, *CCND1* expression was enhanced (11_TRAR), diminished (29_TRAR and 17_TRAR) or remained at similar level (21_TRAR and 28_TRAR) compared with parental cell lines (Figure 1B). Expression of metalloproteinase-2 (MMP-2), an enzyme that participates in increasing melanoma invasiveness during development of resistance [31], was upregulated only in selected resistant cell lines, especially in those derived from DMBC11 drug-naïve cell line (Figure 1B). The mRNA level (Figure 1B) and secretion (Appendix A) of interleukin-8 (IL-8), an important cytokine for melanoma growth and progression [32], were significantly reduced in almost all resistant cell lines. Of note, the IL-8 transcript level was significantly increased in 11_TRAR cells, similarly as for cyclin D1 and MMP-2 (Figure 1B).

### 3.3. Increased Frequency of Nerve Growth Factor Receptor (NGFR)-Positive Cells is Accompanied with Suppression of MITF-Dependent Program in the Majority of Melanoma Cell Lines Resistant to Vemurafenib or Trametinib

As published previously [27], the majority of melanoma cell lines resistant to either vemurafenib or trametinib exerted reduced MITF-M transcript and protein at the bulk levels, and the percentages of MITF-positive cells were also substantially lower in drug-resistant than in drug-naïve cell populations, which is summarized in Figure 1C. This might indicate that other than MITF-dependent program is executed more efficiently in resistant cells. First, we checked whether percentages of neural crest stem-like cells were increased in resistant populations. Indeed, when the level of NGFR, a marker of neural crest-like cells (also known as CD271) was assessed, higher percentages of NGFR-positive cells were detected in the majority of resistant cell populations than in their drug-naïve counterparts (Figure 1C). 

Accession numbers are indicated of resistant cell populations than in their drug-naïve counterparts (Figure 1C). The highest increase was observed in melanoma cell populations resistant to trametinib, with more than 50% of NGFR-positive cells found in the 21_TRAR cell population (Figure 1C). An increase in the percentages of NGFR-positive cells was accompanied with a reduction of the frequencies of MITF-positive cells during development of resistance to vemurafenib or trametinib. Such a phenomenon was observed in 21_TRAR, 21_PLXR, 28_TRAR and 29_PLXR cell populations. In 29_TRAR cells both markers (MITF and NGFR) remained at the similar levels as in drug-naïve cell population, and in 17_TRAR cell population the percentages of NGFR-positive cells were reduced from low (7.5%) to almost undetectable (0.2%) level. In those two cell populations MITF-dependent transcriptional program was not substantially suppressed during development of resistance and they remained MITF^high^/NGFR^low^ [27]. The high percentage of NGFR-positive cells in DMBC11 was almost doubled in 11_PLXR, whereas the percentage of MITF-positive cells remained very low (not shown). Interestingly, in 28_PLXR cell population both markers were reduced, but while MITF-positive cells were almost completely eradicated, the NGFR-positive cells were detectable at 5.8%. Thus, the conclusion that the inverse correlation exists between MITF-positive and NGFR-positive cells cannot be applied for all melanomas developing drug resistance.

### 3.4. AXL is not an Unambiguous Marker of Resistance to Targeted Therapeutics

Next, we checked whether the expression of AXL, a marker of melanoma metastasis recently connected with drug resistance [33,34,35], was also increased in resistant melanoma cell lines obtained in this study. Originally, only two drug-naïve cell lines were AXL^high^ (DMBC12 and DMBC11) (Figure 1D,E). At the transcript level, AXL was reduced or remained unchanged in the majority of resistant cell lines (Figure 1D). The protein level of AXL was not increased in any of AXL^low^ drug-naïve cell line during development of resistance, and in resistant cell lines derived from AXL^high^ cell lines (DMBC11 and DMBC12) was even substantially reduced (Figure 1E). Although drug-naïve melanoma cells were either AXL^high^/MITF^low^ or AXL^low^/MITF^high^, their drug-resistant counterparts while losing expression of MITF did not become AXL^high^. This notion is supported by the analysis of AXL expression using publicly available microarray data (Gene Expression Omnibus (GEO)), which revealed that the development of resistance to vemurafenib or dabrafenib combined with trametinib was accompanied with increased expression of AXL only in selected cases, and the AXL level either remained unchanged or was decreased in the majority of relapse samples (Figure 1F).

### 3.5. Genes from the Sex-Determining Region Y-box (SOX) Family are Differentially Expressed in Resistant and Drug-Naïve Melanoma Cell Lines

In addition to expression of AXL, NGFR and a lineage-specific transcription factor MITF, expression of SOX transcription factors that are associated with control of cancer cell plasticity, is broadly used to classify the phenotypes of melanoma cells [36,37]. First, the expression of *SOX2*, *SOX9,* and *SOX10* at the transcript levels was compared between drug-naïve cell lines. *SOX10* was quite uniformly expressed in all drug-naïve cell lines relative to the median value (Figure 2A). Transcript levels of *SOX2* were below the median value in DMBC21, DMBC28, and DMBC29 cell lines, and markedly higher in DMBC11, DMBC12, and DMBC17 cells (Figure 2A). *SOX9* expression was at a high level only in DMBC33 and DMBC17 cells (Figure 2A). SOX2 transcript levels were significantly elevated in all resistant cell lines, both PLXR and TRAR, compared with the transcript levels in the corresponding drug-naïve cells, except for 17_TRAR cell line (Figure 2B) that originally exerted the highest expression of this gene among drug-naïve cell lines (Figure 2A). Transcript levels of SOX9 and SOX10 were significantly higher in TRAR cell lines compared with the levels in their drug-naïve counterparts, except for *SOX10* expression in 29_TRAR cell line (Figure 2B). In PLXR cells, expression of SOX9 and SOX10 was at the similar levels as in drug-naïve cells, and only SOX9 transcript was significantly lower in 33_PLXR cells and significantly higher in 29_PLXR cells. More pronounced enhancement of *SOX9* expression in TRAR cells than in PLXR cells was clearly visible when gene expression in each resistant cell line was related to the median value (Figure 2B, lower panel). At the protein level, SOX2 level markedly differentiated drug-naïve cell lines from their resistant counterparts, especially PLXR cell lines (Figure 2C). The protein level of SOX10 remained unaltered during development of resistance, except for resistant cells derived from DMBC21 cell line, in which SOX10 protein levels were substantially increased (Figure 2C).

### 3.6. Resistant Melanoma Cell Lines Reactivate the MAPK Signaling Pathway and/or Trigger the Wingless-Type (WNT) Pathway

To examine the dominant signaling cascades during development of resistance to trametinib or vemurafenib, lysates from drug-naïve and drug-resistant melanoma cells were subjected to human phospho-kinase array (Figure 3A). The results indicate that ERK1/2 phosphorylation was elevated in the majority of resistant cells lines. Changes in phosphorylation of other proteins were not massive, and were cell line-specific. 28_PLXR and 28_TRAR cell lines most markedly differed from their parental drug-naïve DMBC28 cell line in the protein phosphorylation status. Besides ERK1/2, also mitogen- and stress-activated protein kinase 1/2 (MSK1/2), ribosomal S6 kinase 1/2/3 (RSK1/2/3), AKT1/2/3, protein kinase with no lysine 1 (WNK1), AMP-activated protein kinase alpha2 (AMPKα2), signal transducer and activator of transcription (STAT6), STAT2 and cAMP responsive element binding protein (CREB) substantially changed their phosphorylation status in these two cell lines compared with other resistant cell lines. As phospho-profiling uncovered variability in activities of the signaling pathways between drug-naïve cell lines, and between drug-naïve cell lines and their drug-resistant counterparts, we assessed phosphorylation status of the main effector proteins by immunoblotting (Figure 3B). Since resistance to inhibitors of the MAPK signaling pathway is frequently associated with the reactivation of this signaling cascade, the levels of activated MEK1/2 (p-MEK1/2) and ERK1/2 (p-ERK1/2) were determined. Changes in phosphorylation of ERK1/2 were not always consistent with alterations of MEK1/2 activity (Figure 3B). While phosphorylation of both, ERK1/2 and MEK1/2 was enhanced in 11_PLXR, 11_TRAR, 21_TRAR, and 28_TRAR cells compared with parental drug-naïve cells or reduced in 12_PLXR and 33_PLXR cells, changes in phosphorylation of ERK1/2 and MEK1/2 in other cell lines were unrelated to each other (Figure 3B). This was clearly pronounced in 21_PLXR and 28_PLXR cells, but also in 29_TRAR cells. In 17_TRAR cells, p-MEK1/2 remained almost undetectable and the level of p-ERK1/2 was diminished. The WNT pathway activity, assessed as the level of active β-catenin, was enhanced in the majority of resistant cell lines compared with their drug-naïve counterparts (Figure 3B).

Searching for mechanisms underlying resistance, we found that the levels of BRAF protein (Figure 3B) and transcript (Figure 3C) were markedly higher in 11_TRAR cells compared with parental drug-naïve cell line. Comparison of the expression of BRAF to the median value of all resistant cell lines revealed that 28_PLXR exerted the lowest expression of BRAF, which did not correspond to one of the highest increase in activity of ERK1/2. Interestingly, dual specificity MAPK phosphatase 6 (DUSP6), known to dephosphorylate ERK1/2, was either eradicated or remained almost undetectable in resistant cell lines (Figure 3B), which might contribute to increased ERK1/2 phosphorylation in e.g., 28_PLXR cell line. Acquired genetic alterations were not found in *DUSP6* in any resistant cell line (Appendix A). To delineate possible mechanisms responsible for observed changes in activity of the MAPK signaling pathway, other genetic alterations acquired in resistant cell lines were analyzed. When genes previously associated with resistance to inhibitors of the MAPK signaling pathway were analyzed, not many SNPs and InDels were found (Figure 3D, Appendix A). Alterations of *NRAS* were not detected (Appendix A). Nonsense variants at E69 and E61 of *CDKN2A* were gained by 21_PLXR, 28_PLXR, and 33_PLXR cells (Figure 3D). Loss of heterozygosity of *BRAF*^V600E^ (Figure 3D) in PLXR cell lines could not explain all changes in expression of *BRAF* and activities of MEK1/2 and ERK1/2. Mutations with damaging Polyphen-2 predictions that lead to the E637K substitution in fibroblast growth factor receptor 2 (FGFR2), and several substitutions in RAS guanyl-releasing protein 4 (RASGRP4) and son of sevenless homolog 1 (SOS1) were found in 21_PLXR and 28_PLXR cells (Figure 3D). In 28_PLXR cells, a D201V substitution in Moloney murine sarcoma viral oncogene homolog (MOS) and L437F and K438M substitutions in erythroblastic leukemia viral oncogene homolog 4 (ERBB4) as well as a disruptive in-frame insertion in the R426 position of ERBB4 were found in addition (Figure 3D) providing a possible explanation for enhanced ERK1/2 activity in this cell line compared with drug-naïve cells. In 11_PLXR cells, frameshift variants of ERBB4 were gained. No new mutations in the MAPK signaling pathway-related genes were acquired in 29_PLXR and 12_PLXR cells (Figure 3D, Appendix A). In melanoma cells resistant to trametinib, few damaging mutations were found including L437F and K438M substitutions in ERBB4 in 11_TRAR, G388R variant of FGFR4, and E36A variant of platelet-derived growth factor receptor alpha (PDGFRA) in 21_TRAR, and E519K alteration of RASGRP4 in 17_TRAR cells. In contrast to 28_PLXR cells, no acquired mutations in genes related to the MAPK pathway were found in 28_TRAR cell line (Figure 3D). In 29_TRAR cell line, exerting undetectable level of p-MEK1/2, de novo heterozygous mutations in MAP2K2 giving rise to MEK2^L201V^ and MEK2^F57V^ variants were found (Figure 3D). Low level of MEK1/2 in 17_TRAR cells might be a consequence of the mutation leading to MEK1^P124S^ variant, already present in DMBC17 drug-naïve cells [23].

### 3.7. Genetic Analysis Revealed RNA-Binding Motifs X (RBMX) as a Putative Gene Associated with Acquired Resistance of Melanoma Cells with Enhanced Activity of the PI3K/AKT Signaling Pathway

Concerning the PI3K/AKT signaling pathway, low level of phospho-AKT remained unchanged in 17_TRAR and 29_TRAR cells, two resistant cell lines with differentiation-related phenotype. In phospho-AKT^high^ drug-naïve cell lines, activity of AKT remained unchanged (11_PLXR cells) or was substantially diminished (11_TRAR and 12_PLXR cells) during the development of resistance (Figure 3B). AKT activity was the most substantially enhanced in 29_PLXR, 21_TRAR, and 28_TRAR cell lines compared with their drug-naïve counterparts, but also raised in 21_PLXR and 33_PLXR cells (Figure 3B). Several potentially damaging variants of RBMX, including one frameshift variant (S303fs) were acquired in melanoma cells with strongly enhanced activity of AKT (29_PLXR, 21_TRAR, and 28_TRAR cells). As similar alterations of RBMX, except for the S303fs variant were also found in 29_TRAR cells, in which phosphorylation of AKT was not enhanced, it could be speculated that only S303fs variant of RBMX contributes to the increased activity of AKT, which should be confirmed experimentally.

### 3.8. Resistant Cell Lines do not Frequently Acquire Growth Factor Dependence, and Epidermal Growth Factor (EGF) is Indispensable for Survival of Resistant Cells Only in the Presence of Drug

We have already reported that melanoma cells do not depend on exogenously provided growth factors, EGF, basic fibroblast growth factor (bFGF) and hepatocyte growth factor (HGF), and lack of them in the culture medium do not affect cell morphology and activity of several signaling pathways in patient-derived melanoma cells [25]. In this study, analysis of genetic material from resistant cells revealed several acquired mutations in genes encoding receptors of growth factors, especially FGFR (Figure 3D). To investigate growth factor dependence of PLXR and TRAR cell lines, selected resistant cells were grown either without exogenously added growth factors (noGF) or in the presence of EGF, bFGF, and HGF provided as single factors or in combination. Although de novo mutations in the gene encoding epidermal growth factor receptor (EGFR) that lead to a constitutive activity of the corresponding receptor were not found in resistant cells (Appendix A), a phospho-protein array revealed a substantial increase in the level of active phospho-EGFR in 28_TRAR cells compared with matched drug-naïve cells treated with 50 nM trametinib (Figure 3A and Figure 4A). To check whether this phenomenon is solely related to the receptor activation or it is also associated with different expression of *EGFR*, the protein level of EGFR was assessed by immunoblotting. All drug-naïve cell lines were characterized by almost undetectable *EGFR* expression. EGFR protein level was the most markedly increased in 28_TRAR cells (Figure 4B), which indicates that both, expression and activity of EGFR were enhanced in this cell line during the development of resistance. Proliferation curves of PLXR cell lines were overlapping when comparing cell behavior in different growth factor cocktails, including noGF conditions (Figure 4C). TRAR cell lines exerted more variable response to growth factors. 29_TRAR cells (Figure 4C) and 17_TRAR cells (not shown) did not develop dependence on any growth factor. A slight influence of lack of exogenously added EGF on proliferation of 21_TRAR cells was visible after 72 h (Figure 4C). In contrast, lack of EGF in the culture medium resulted in substantial reduction of a viable cell number in 28_TRAR cell line already after 48 h (Figure 4C).

To determine the molecular alterations associated with the response of resistant cell lines to growth factors, 28_TRAR cells were grown for 19 h in a standard EGF/bFGF-containing medium, and in media supplemented with either EGF, bFGF, or HGF, or without growth factors (noGF). 21_PLXR, 21_TRAR, and 28_PLXR cells were used for comparison. The activity of EGFR downstream signaling pathways remained unaffected in EGFR^low^ resistant cell lines, 21_PLXR and 28_PLXR (Figure 4D). In contrast, lack of EGF in the culture medium resulted in markedly reduced activity of AKT, MEK1/2, and ERK1/2 in 21_TRAR cells, and complete inhibition of activities of these kinases in 28_TRAR cells (Figure 4D). Notably, EGFR protein levels were diminished in both TRAR cell lines deprived of EGF stimulation, however, more dramatic change was observed in 28_TRAR cells (Figure 4D). Furthermore, downregulation of SOX10 protein level and induction of apoptosis assessed as cleavage of PARP after 19 h were substantial in 28_TRAR cells grown in the absence of EGF, and these changes were not found in 21_TRAR cells (Figure 4D). Results shown in Figure 4A–D indicate that EGFR activation occurred in a ligand-dependent manner. Next, we asked the question whether EGFR activation is also necessary for maintenance of 21_TRAR and 28_TRAR cell lines in the absence of trametinib. A removal of the drug from the culture medium resulted in EGF-independent proliferation of both cell lines, which was especially surprising for 28_TRAR cells (Figure 4E).

### 3.9. While Trametinib Inhibits Elevated ERK1/2 Activity and Proliferation of PLXR Cells, Cross-Resistance to Vemurafenib is Developed in TRAR Cell Lines

As both, vemurafenib and trametinib act within the same signaling pathway, we assessed how melanoma cell lines that are resistant to one of these inhibitors respond to complementary one. Levels of phosphorylated MEK1/2 and ERK1/2 were determined after 19 h treatment with a complementary drug (Figure 5A). Targeting MEK1/2 with trametinib in PLXR cell lines resulted in the accumulation of MEK1/2 phosphorylated on Ser^217/221^ (Figure 5A). Notably, this effect was concomitant with a consistent lack of ERK1/2 activity (Figure 5A). In TRAR cell lines, vemurafenib reduced levels of p-MEK1/2, whereas levels of p-ERK1/2 remained unaltered, except for 11_TRAR and 17_TRAR cells. Levels of p-MEK1/2 and p-ERK1/2 were elevated by vemurafenib in 17_TRAR cells (*BRAF*^wt^/*HRAS*^Q61R^). Lack of cleaved PARP indicates that apoptosis was not induced in any of cell lines (Figure 5A), which together with short incubation time suggest that observed changes in the MAPK pathway were rather due to cell reprogramming than a cell selection process.

To evaluate the consequences of observed alterations in the MAPK pathway, TRAR cells were treated with vemurafenib and PLXR cells were treated with trametinib for up to three days and viable cell numbers were assessed (Figure 5B). Trametinib reduced a number of viable cells in PLXR cell lines, except for 12_PLXR cell line. Notably, strong inhibitory effects were found in 11_PLXR and 28_PLXR cells, even if the concentration of trametinib was reduced from 50 nM to 20 nM (Figure 5B). In contrast, TRAR cell lines were insensitive to BRAF^V600E^ inhibition independently of vemurafenib concentration used in the study (Figure 5B). In general, inhibition of proliferation was substantial in these resistant cell lines that exerted markedly reduced ERK1/2 activities. In addition, trametinib-triggered strong inhibition of proliferation in 11_PLXR and 28_PLXR cell lines could be explained by a significant reduction of *CCND1* expression as assessed at the transcript level (Figure 5C). Notably, short drug discontinuation (short drug holiday) did not substantially changed proliferation curves of any of four representative cell lines when compared with curves of drug-treated resistant cells (Figure 5D). This might indicate that the response of vemurafenib-resistant cells to trametinib (Figure 5B) was rather a consequence of the presence of trametinib than vemurafenib withdrawal. In addition, viability of 29_PLXR, 28_TRAR and 29_TRAR cells remained unaffected when drug holiday was extended to 10 days (Figure 5D). Although proliferation of 28_PLXR cells on long drug holiday was slower, a time-dependent increase in viable cell number was still observed (Figure 5D).

### 3.10. Several Alterations in Signaling Pathways Observed in Drug-Resistant Cells are Already Induced during Immediate Response to Drugs

We asked the question, which of the observed alterations that differentiate drug-naïve and drug-resistant cells could be associated with the development of resistance, and which with immediate response to drugs. For that additional experiments analyzing difference between melanoma cells permanently treated with drugs (resistant cells) and drug-naïve cells after short treatment (immediate response) were performed. Gene expression at the transcript and protein levels as well as activity of corresponding proteins were (re)-analyzed based on results of the present report and previously published data [22,27] and shown in Figure 6. First, lysates from shortly (19 h) exposed to drugs melanoma cells, either drug-naïve or drug-resistant, were subjected to human phospho-kinase array (Figure 6A).

Phosphorylation status of the majority of proteins was similar in both settings (Figure 6A). These results combined with the results of immunoblotting (Figure 2C, Figure 3B and Figure 6B [27]) demonstrated that only a few proteins exerted different phosphorylation status or were differentially expressed, as summarized in Figure 6C. ERK1/2 phosphorylation while suppressed by short treatment [27], was elevated in several of drug-resistant cell lines (Figure 6C). Phosphorylation status of AKT^S473^ also diverse in drug-naïve cells (Figure 6B), mostly remained unaltered during immediate response, but was substantially changed in the majority of drug-resistant cell lines in cell line- and drug-dependent manner (Figure 6C). Phosphorylation of MEK1/2, which was reduced by short exposure to vemurafenib or trametinib [27], was also reduced in the majority of resistant cell lines, however, an opposite effect was also found in selected, especially TRAR cell lines (Figure 6C). β-catenin activity was enhanced in almost all resistant cell lines, whereas after short treatment this activity was mostly unaltered (Figure 6B,C). In summary, resistant cells responded to drugs with different activities of ERK1/2, AKT and β-catenin than their drug-naïve counterparts, whereas MEK1/2 activity was differentially changed almost exclusively in TRAR cell lines and *SOX2* expression was differentially changed only in PLXR cell lines compared to their matched drug-naïve cell lines. *DUSP6* expression was reduced already during short treatment, and these effects were kept at the similar level in the resistant cell lines. No effects were found for SOX10, except for 21_PLXR and 21_TRAR cell lines, in which SOX10 was markedly elevated (Figure 6C). *AXL* expression remained undetectable in the majority of cell lines, however, in AXL^high^ drug-naïve cells short treatment slightly induced expression of AXL, whereas development of resistance reduced the level of this protein. During development of resistance to vemurafenib or trametinib, NGFR signaling was either significantly suppressed (28_PLXR and 17_TRAR cells), remained unchanged (29_TRAR) or was significantly enhanced (all other resistant cell lines), whereas during immediate response the percentages of NGFR-positive cells were increased in all melanoma cell lines although to different extent (Figure 6C).

## 4. Discussion

A melanoma relapse is observed in the majority of patients treated with targeted therapeutics, irrespectively of the initial response [5,38]. In our in vitro study most drug-naïve melanoma cell lines derived from the tumor specimens also developed resistance after 4–5 months of constant exposure to drugs. Analysis of whole-exome sequencing data revealed only a few mutations previously identified as associated with acquired resistance to targeted drugs. It is in line with the current view that in parallel to genetic alterations, melanoma cell-autonomous mechanisms of resistance can rely on complex transcriptomic adaptations supported by epigenetic regulations, including miRNA-mediated mechanisms, and extended by the interplay between tumor and stromal cells that involves cell-cell interactions, paracrine stimulation and extracellular vesicles to create a resistance-permissive niche [39,40,41,42,43,44,45,46]. Mechanisms of resistance primarily directed to support proliferation and survival of cancer cells implicate a plethora of transcriptional regulators, adaptive responses, and feedback loops that extensively affect melanoma cell phenotype enabling the expansion of distinct cell subpopulations in the presence of drug(s) [27,47,48,49,50,51,52]. Several evidence indicates that a multicellular heterogenous ecosystem of melanoma is changed in a stepwise manner during development of resistance, and many alterations accompanying this process are reversible in nature, which contributes to a high phenotypic plasticity of melanoma cells [16,36,53,54,55,56,57]. Diverse subpopulations within the tumor are considered as key contributors to variable inter-patient responses to therapy [58], although relapsed melanomas at distinct sites of the same patient can also rely on different resistance programs [12].

The plasticity of melanoma as enabler of drug resistance was a subject of most recent reviews [16,37,53]. Arozarena and Wellbrock [53] proposed three phases of response to MAPK inhibitor therapy in MITF^high^ melanomas, initial with induced reprogramming, adaptive with enhanced drug tolerance and acquired with drug resistance. Based on the presented model, inter-patient variability is observed already in the initial response to treatment as three different dominating phenotypes can be generated: MITF^high^ (pigmented/proliferating), AXL^high^ (invasive), and NGFR^high^ (primitive, neural-crest stem-like). In acquired resistance phase, about 70% of melanomas present the AXL^high^ phenotype and only about 23% are enriched with the MITF^high^ phenotype. Our present study as well as previously published [27], which brought together phenotypic changes in the initial phase of response and genetic/phenotypic alterations in the acquired drug-resistant phase, strongly support this view although we are far in our opinions from making any general statement about predominant transcription programs that govern the development of drug resistance in melanoma. On the contrary, we did observe a high variability and plasticity in execution of the diverse programs between different melanoma cell lines that developed resistance to either vemurafenib or trametinib. Drug-naïve melanoma cell lines derived from different tumor specimens used in our study to obtain resistant cells were either MITF^high^/AXL^low^ (five cell lines) or MITF^low^/AXL^high^ (two cell lines). Among eleven resistant cell lines, six vemurafenib-resistant and five trametinib-resistant, none became MITF^low^/AXL^high^ (this study and [27]). Thus, while drug-naïve cell lines consistently followed the mutually exclusive AXL and MITF programs, a switch towards AXL-dependent cell state during development of resistance is not unambiguous. Therefore, our results do not support the previous findings that AXL can be considered as a biomarker of resistance [33,34,35]. This conclusion although based on in vitro study is also supported by the analysis of publicly available microarray data (Gene Expression Omnibus (GEO)) showing that the substantial enhancement of the AXL expression occurred only in the minority of relapse samples. Our results are also contradictory to the statement that AXL-negative cells are sensitive to inhibition of the MAPK pathway [33,34,35], as all resistant cell lines originally being or becoming AXL^low/no^ could continuously grow in the presence of vemurafenib or trametinib in the resistant phase. Enhanced expression of NGFR was shown in melanoma cells with acquired resistance to BRAF inhibition or combined BRAF/MEK inhibition [59,60], but it was also a part of a response to treatment preceding development of resistance to MEK inhibition [61]. Our study indicates that while the percentages of NGFR-positive cells were uniformly increased after short treatment with vemurafenib or trametinib, not all melanoma cell populations resistant to either of these drugs showed higher frequency of NGFR-positive cells than their drug-naïve counterparts. As NGFR can trigger signaling in the absence of ligand, act as a homodimer or interact with the large number of different partners [62,63], it can be a component of diverse and possibly opposing signaling pathways and contribute to phenotype switching in melanoma [64]. Therefore, different expression of both, NGFR and MITF in melanoma cells might be responsible for diverse outcomes at the level of phenotype. In summary, as we obtained several phenotypes when only three markers were considered, e.g., MITF^high^/AXL^low^/NGFR^low^ (17_TRAR, 29_TRAR), MITF^low^/AXL^low^/NGFR^low^ (28_PLXR, 21_PLXR), MITF^low^/AXL^low^/NGFR^high^ (11_PLXR), MITF^medium^/AXL^low^/NGFR^high^ (21_TRAR, 28_TRAR), patient-to-patient variability might be richer and more nuanced than previously suggested. Especially interesting is MITF^low^/AXL^low^/NGFR^low^ phenotype of 28_PLXR cell line. This is vemurafenib-resistant cell line, in which dedifferentiation, shown as inhibition of expression of MITF and MITF-dependent pigmentation-related genes, is irreversible [27]. In the absence of MITF that is known to mediate survival [65,66,67], diverse receptor tyrosinase kinases are thought to supply MAPK-independent pro-survival signals [11,35,68]. In 28_PLXR cells neither AXL nor NGFR were activated, however, there was a combination of several mutations in *FGFR*, FMS-like tyrosine kinase 3 (*FLT-3*) and *ERBB4*, which is exclusively observed in this cell line. Moreover, enhanced activity of STAT3 followed by increased expression of *SOX2*, which were observed only in 28_PLXR cells could be also considered as the mechanism supporting cell survival. It has been demonstrated that SOX2 as a part of STAT3-SOX2-CD24 axis rescued melanoma cells against acute exposure to vemurafenib [69]. SOX2 was shown to contribute to the stable reprogramming of melanoma cells to their undifferentiated counterparts that were resistant to inhibitors of the MAPK signaling pathway independently of which of oncogenic mutations in the MAPK pathway (*BRAF* or *NRAS*) was present [70], and SOX2-positive cells were reported in melanoma cell populations exerting a drug-tolerant state to combined dabrafenib and trametinib treatment [71]. Reciprocally excluding SOX2 and MITF expression was reported [72], which was also detected in all vemurafenib-resistant cell lines investigated in our study. Changes in *SOX2* expression in 29_TRAR and 17_TRAR cells were less pronounced probably because of an elevated MITF level and high frequency of differentiated cells [27]. SOX2 level in 17_TRAR cells might also be reduced because of possibly damaging mutation T222I+/−. Similarly as previously shown [73], vemurafenib resistance was developed independently from SOX2 expression in drug-naïve cells, however, while in the study of Cesarini et al., SOX2 was strongly upregulated only in one out of three vemurafenib-resistant cell lines, in our study SOX2 upregulation was observed in all six PLXR cell lines. As several reports suggest genetic context-dependent role of SOX2 in melanoma [73,74,75,76,77], regulation of SOX2 expression and the role of this transcription factor in resistant melanoma cells remain to be clarified.

Proliferation curves of resistant cells and their drug-naïve counterparts were similar. While immediate response to PLX and TRA was associated with reduced cyclin D1 transcript levels in all drug-naïve cell lines [22], variable alterations in *CCND1* expression, including upregulation, was found in resistant cells. The stop-gained variants of *CDKN2A* were acquired by two vemurafenib-resistant cell lines, 21_PLXR and 28_PLXR, and a possibly damaging D148N variant of E2F transcription factor 3 (E2F3) was additionally found in 21_PLXR cells (Appendix A), but the level of cyclin D1 in these cell lines and many other resistant cell lines was not elevated, which suggests that the sustained proliferation is caused by different mechanism(s). On the contrary, proliferation of 11_TRAR cells may depend on stop-gained variants of *CDKN2A* found already in matched drug-naïve cell line, and a significant upregulation of BRAF and cyclin D1, which might be the result of amplification of *BRAF* [9] and *CCND1* [78], respectively. Both aberrations have been previously identified as the resistance mechanisms, and co-existence of *CDKN2A* nonsense variants with elevated level of cyclin D1 can amplify signaling through cyclin-dependent kinase 4 (CDK4) [79]. Interestingly, elevated BRAF expression, MEK1/2 activity and cyclin D1 transcript level in 11_TRAR cells were accompanied with increased level of IL-8. This cytokine has been already implicated in promoting tolerance of melanoma cells to BRAF inhibition, and its receptor C-X-C motif chemokine receptor 2 (CXCR2) was suggested as a potential target in melanoma [80]. In the study of Young et al. [80], IL-8 that stimulated the survival of melanoma cells in the presence of BRAF and MEK inhibitors, was released by fibroblasts located in inflammatory niche of the tumor, whereas our study suggests that autocrine signaling is also possible.

Reactivation of the MAPK (ERK1/2) pathway [5,81,82] or compensatory activation of the PI3K/AKT pathway [83] are the most common mechanisms of drug resistance, however, two resistant cell lines, 12_PLXR and 17_TRAR, could be propagated in the presence of respective drugs even if the MAPK (ERK1/2) and PI3K/AKT pathways were not activated. A substantial downregulation of phospho-ERK1/2 without accompanying activation of AKT has been already shown in resistant melanoma cells [84]. Elevated activity of ERK1/2 was observed in 7 out of 11 resistant cell lines, and interestingly this was accompanied by increased phosphorylation of MEK1/2 only in three resistant cell lines. Several mutations in the MAPK cascade-related genes were found (Appendix A), including two missense variants (F57V and L201V) of MEK2 in 29_TRAR cells. It has been demonstrated that MEK variants can contribute to resistant phenotype of melanoma cells, as (i) cells harboring MEK2^F57C^ substitution exerted phospho-MEK2^low^/phospho-ERK1/2^high^ signature [85], (ii) regrowth of melanoma after initial response to MEK inhibitor has been attributed to MEK1^P124L^ variant [86], and (iii) untreated patients harboring MEK1^P124S^ or MEK1^P124L^ variants faced rapid progression upon administration of BRAF inhibitor [13]. Another way to keep ERK1/2 active is to alter the activity of DUSP6, a specific phosphatase of ERK1/2 [87,88]. While inverse association between the levels of p-ERK1/2 and DUSP6 was found in drug-naïve cells [23], DUSP6 suppression in resistant cells was not consistently associated with an increase in phospho-ERK1/2 level, suggesting that downregulation of DUSP6 was not the only mechanism reactivating ERK1/2. Moreover, DUSP6 level was reduced already after short treatment of melanoma cells with either vemurafenib or trametinib, which further supports the notion that DUSP6 suppression might not be essential for the development of resistance to targeted therapeutic. None of (epi)genetic mechanisms that might influence a DUSP6 level [89,90] have been detected in our study. Moreover, assessment of pharmacodynamics effects of binimetinib (an inhibitor of MEK1/2) on MAPK signaling revealed no association of between the DUSP6 level/phosphorylation of ERK1/2 and clinical efficacy [91].

The activity of AKT signaling pathway was originally high in two drug-naïve cell lines and was elevated in a number of resistant cell lines. Interestingly, we have shown that activation of AKT in resistant cell lines might be connected with acquired mutations in a gene encoding RBMX, an RNA-binding protein protecting cells against DNA damage and acting as a tumor suppressor [92,93]. Mutations in RBMX were recently reported in papillary thyroid carcinoma cells resistant to vemurafenib [94], and knock-down of wild-type RBMX caused an increase of p-AKT protein level in naïve cancer cells. Several de novo mutations (G379R, Y357H, R339G, S337N, S303 frameshift) in RBMX that span the RNA-interacting domain of RBMX [92] were found in our study, but only one of them S303fs+/− was present in resistant melanoma cell lines with elevated activity of AKT (21_TRAR, 28_TRAR, 29_PLXR). The other four mutations were also acquired by 29_TRAR, but enhanced activity of AKT was not detected. The role of RBMX protein in inhibiting the PI3K/AKT pathway in melanoma, especially the functional inability of a frameshift variant S303 of RBMX to exert this function, needs a further investigation. Downregulation of AKT activity could be connected with loss of AXL in 11_TRAR and 12_PLXR cell lines, as positive regulation between AXL and AKT has been reported [95]. An elevated level of SOX10 protein in resistant cell lines derived from DMBC21 cells might contribute to enhanced activation of AKT [96]. And finally, enhanced activation of AKT might also result from the relief of ERK-dependent negative feedback as it was probably a case in 33_PLXR cells, in which all components of the BRAF/MEK/ERK pathway were downregulated.

Another important issue highlighted in our study is remarkable plasticity of resistant melanoma cells. This was visible in their response to changes in the microenvironment, introduced either as a treatment with a complementary drug or as different composition of growth factors. In the cross-resistance experiments, substantial alterations of ERK1/2 and MEK1/2 activities could be induced within 19 h in resistant cell lines. All four PLX-resistant p-ERK^high^ cell lines were responsive to trametinib as evidenced by reduction of the proliferation rate and inhibition of ERK1/2 activity. Surprisingly, this was accompanied by strong enhancement of MEK1/2 phosphorylation. The accumulation of phosphorylated MEK1/2 but downregulation of ERK1/2 activity suggest that trametinib while leading to hyperactivation of positive regulators upstream of MEK1/2, still efficiently inhibits kinase activity of MEK1/2. Accordingly, a paradoxical activation of MEK1/2 following treatment with trametinib has been reported in resistant cells harboring *BRAF* amplification and MEK2^Q60P^ variant [97]. On the contrary to PLXR cell lines, TRA-resistant cell lines developed cross-resistance to vemurafenib. The majority of TRAR cell lines could preserve the ERK1/2 activity and uniform unresponsiveness to vemurafenib shown as lack of changes in the proliferation rate. Cross-resistance of 29_TRAR cell lines might be explained by the presence of MEK2^F57V^ variant. The variant MEK2^F57C^ has been reported as being partially RAF-independent and insensitive to BRAF inhibition [85]. In other cell lines, 21_TRAR and 28_TRAR, sustained ERK1/2 activity despite exposure to vemurafenib could be associated with high activity of AKT pathway [98], justifying vemurafenib-triggered changes of ERK1/2 phosphorylation in phospho-AKT^low^ 11_TRAR cells. Cross-resistance has been inconsequently associated with the acquisition of NRAS^Q61K^ variant in preclinical settings [99,100] and loss of *MITF* expression [35]. The limitation of our approach is that while it enables to draw additional conclusions concerning mechanisms of the MAPK cascade reactivation, these molecular and cellular effects were assessed only for up to three days, whereas long-term clinical observations indicate weak response to trametinib administered sequentially to patients previously treated with BRAF inhibitor [101]. Combined treatment with BRAF+MEK inhibitors can prolong clinical response [3,4], but acquired resistance also inevitably occurs, and usually follows the augmentation or combination of mechanisms of resistance to a single drug [81].

A high plasticity of resistant melanoma cells was also detected as the rapid regrowth and reprogramming of EGF-dependent EGFR^high^ trametinib-resistant cells subjected to drug holiday. While elevated EGFR expression has been previously found in melanomas resistant to vemurafenib [102,103,104,105], for the first time we have shown that an enhancement of EGFR expression and activity can be associated with the development of resistance to trametinib. Moreover, the level of EGFR protein, which was medium in 21_TRAR cells and very high in 28_TRAR cells, reflected the extent of EGF dependence. Several mechanisms have been implicated in the regulation of *EGFR* expression including suppression of SOX10 [102,103] followed by upregulation of neuropilin-1 (NRP-1)-c-Jun N-terminal kinase (JNK)-SOX2 axis [106], and SOX9 expression [107]. It is unlikely that these mechanisms are involved in the regulation of EGFR in resistant cell lines investigated in our study as (i) changes in SOX9 and SOX10 expression were not uniformly correlated with EGFR protein level, (ii) EGF deprivation was associated with downregulation of SOX10 in EGFR^high^ 28_TRAR cells suggesting even a concurrent regulation, and (iii) TRAR cells neither expressed SOX2 protein at the detectable level nor activated JNK as shown by the phospho-protein array. In addition, mutually exclusive expression of MITF and EGFR was reported [35,102], and indeed some EGFR-expressing trametinib-resistant cell lines can be characterized as MITF^low^ [27], however MITF loss in PLXR cells was not followed by increased expression of EGFR. Suppression of MITF was also associated with increased expression of ligands synergizing with EGFR activation [102], and autocrine EGF-EGFR circuit was observed in the cultures of resistant melanoma cell lines [108]. In addition, EGF was shown to both activate the receptor and promote its expression [109]. Thus, as EGF deprivation resulted in a marked decrease of EGFR protein level, it can be assumed that EGF sustained *EGFR* expression in both 21_TRAR and 28_TRAR cells. Moreover, as deprivation of exogenous EGF was enough to trigger inhibition of EGFR signaling, production of EGFR-activating ligands in resistant melanoma cells was insufficient. Addiction to EGFR activation was, however, advantageous to melanoma cells only in the presence of drug since the cessation of trametinib treatment led to a growth factor-independent maintenance of TRA-resistant EGFR-expressing cells. This was an immediate effect, which was presumably due to a high plasticity of melanoma cells, and not as a result of depletion of EGFR^high^ subpopulation as reported in BRAF inhibitor-resistant cells [103]. *EGFR* is expressed in epithelial cells and its overexpression is found in cancers of epithelial origin [110], whereas melanoma cells rarely express *EGFR* [111]. Positive staining of EGFR was observed only in 13 out of 114 melanoma samples (11.4%), and the staining intensity was predominantly weak to moderate [112]. *EGFR* expression is more frequent in metastatic melanoma [110] and has been associated with undifferentiated phenotype of melanoma cells [36]. In addition, soluble extracellular domain of EGFR that retains ligand-binding capacity and may dimerize with membrane-bound EGFR to inhibit its activity shows gradual decrease with the increasing stage of the disease [113]. Short treatment of melanoma cells with BRAF inhibitors has led to either augmented EGFR protein level as the result of epigenetic regulation [104] or increased phosphorylation of the baseline EGFR [105]. In addition, EGFR was correlated with resistance to BRAF inhibitors [102,103] by contributing to activity of ERK1/2 and AKT through a physical interaction with urokinase type plasminogen activator receptor (uPAR) in melanoma cells [114,115]. Our study has shown that selected trametinib-resistant cells can adaptively use EGFR, thus providing a therapeutic window for EGFR inhibition as a complementary treatment approach.

## 5. Conclusions

This in vitro study was performed to characterize melanoma cells that developed resistance to vemurafenib or trametinib after 4–5 months of treatment. A high variability of molecular and cellular changes was observed, as there was no similar pattern of genetic/non-genetic alterations among eleven investigated resistant cell lines, including phenotypic changes in expression/activity of crucial melanoma regulators e.g., MITF, AXL, NGFR, AKT, ERK1/2, and SOX2 (Figure 7 and Appendix A). Moreover, vemurafenib- and trametinib-resistant cells differed even if they were obtained from the same drug-naïve melanoma cell line. This means that most of genetic and non-genetic alterations are generated in a cell line- and/or drug-specific manner during long exposure to drugs. Moreover, our study indicates that (1) AXL is not an unambiguous marker of resistance to targeted therapeutics; (2) the development of resistance is accompanied by increased percentages of NGFR-positive cells and reduced percentages of MITF-positive cells in the majority but not all melanoma cell populations. Therefore, the results showing a positive or negative alignment of expression or activity of two or more proteins should be interpreted cautiously as they might apply only to selected patients. Several changes previously assigned to the development of resistance were induced already by short treatment to the extent measurable at the bulk levels, which means that these alterations observed during initial response do not affect only a small subpopulation of melanoma cells. Results presented in this report indicate that melanoma cells, both drug-naïve and drug-resistant present remarkable plasticity. EGF-dependence was limited to the trametinib-resistant cell line with elevated level of EGFR expression/phosphorylation, and it was conditional as excluding trametinib from the culture medium lacking EGF resulted in the rapid regrowth and reprogramming of melanoma cells. Alterations observed in the cross-resistance experiments, best exemplified by phosphorylation of MEK1/2 in vemurafenib-resistant cells after short exposure to trametinib, also indicate remarkable plasticity of resistant melanoma cells. Whole-exome sequencing revealed mutations, previously found in preclinical studies and in tumor samples from patients developing drug resistance. For the first time, we have reported a frameshift variant of RBMX in melanoma. This alteration was present exclusively in resistant cell lines with acquired elevated activity of AKT, which should be further investigated as a potential mutation-based route of melanoma resistance to targeted therapeutics. Considering genetic and phenotypic heterogeneity of resistant melanomas, innovative diagnostic and therapeutic approaches are still needed to forestall the emergence of resistance and improve durability of response. As suggested in recently published review by Boumahdi and de Sauvage [16], targeting cancer cell plasticity should provide an opportunity to increase the efficacy of anticancer treatment, however, a comprehensive landscape of cancer cell plasticity still remains to be established.

## Figures and Tables

**Figure 1 cells-09-00142-f001:**
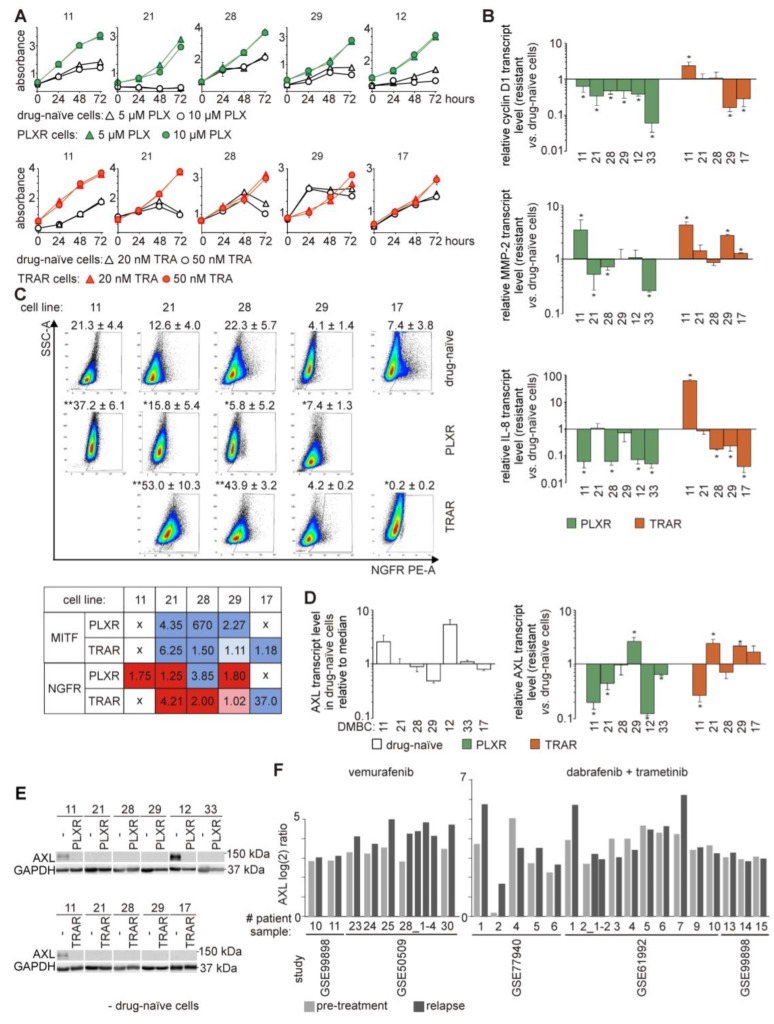
Melanoma cell lines resistant to vemurafenib (PLXR) or trametinib (TRAR) differ from their drug-naïve counterparts. (**A**) Proliferation curves of drug-naïve, PLXR and TRAR cells grown in the presence of drugs at indicated concentrations based on the activity of acid phosphatase. Representative results of triplicate experiment are shown; *n* = 3, except for 12_PLXR cells (*n* = 2). (**B**) Transcript levels of cyclin D1, MMP-2 and IL-8 assessed using qRT-PCR. Bars represent mean values of 3 biological replicates ±SD, except for 12_PLXR cells (*n* = 2). Differences are considered significant at * *p* < 0.05. (**C**) The percentages of nerve growth factor receptor (NGFR)-positive cells are increased while MITF-positive cells are decreased in the majority of melanoma cell populations. NGFR-positive cells are shown as representative density plots, and average percentages ±SD are shown. * *p* < 0.05, ** *p* < 0.01. Comparison of changes in the percentages of MITF-positive and NGFR-positive cells assessed by flow cytometry is shown in the table. The original data for MITF was published elsewhere [27]. The numbers indicate fold change, drug-resistant vs. drug-naïve cell populations. The results were color coded (red, fold increase; blue, fold decrease). (**D**) AXL transcript levels assessed using qRT-PCR. Left: transcript levels of AXL determined in drug-naïve cells relative to the median value in all seven cell lines. Right: expression of AXL assessed in PLXR and TRAR cell lines relative to its expression in matched drug-naïve cells. Bars represent mean values of 3 biological replicates ±SD, except for 12_PLXR cells (*n* = 2). Differences are considered significant at * *p* < 0.05. (**E**) Representative Western blot images showing AXL levels in drug-naïve and drug-resistant cell lines. GAPDH was used as a loading control. *n* = 2. (**F**) Analysis of AXL expression in tumor specimens collected from patients before treatment and post-relapse with resistance developed either to vemurafenib or combined treatment, dabrafenib and trametinib. Results are shown as log 2 ratios normalized to the mean intensity of pre-treatment specimens. Data were obtained from NCBI GEO (http://www.ncbi.nlm.nih.gov/geo/). Accession numbers are indicated.

**Figure 2 cells-09-00142-f002:**
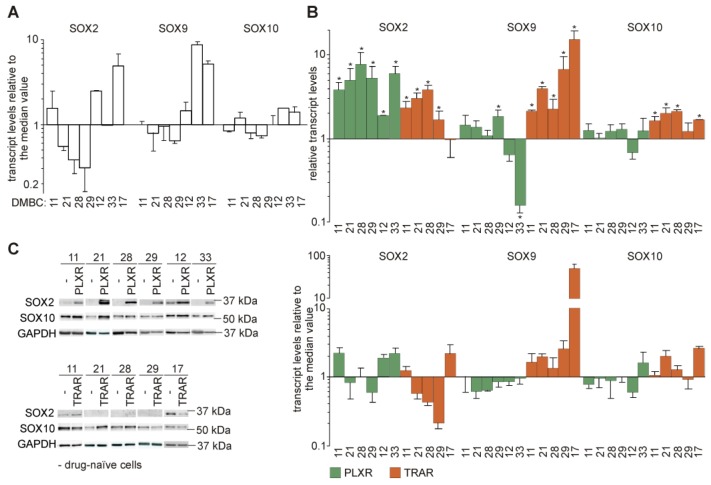
Drug-naïve and their drug-resistant counterparts differ in expression of genes of the SOX family. (**A**) Transcript levels of indicated genes determined in drug-naïve cells by qRT-PCR are shown relative to the median value in all seven cell lines. (**B**) Upper part: expression of *SOX2*, *SOX9* and *SOX10* assessed in vemurafenib-resistant (PLXR) and trametinib-resistant (TRAR) cell lines by qRT-PCR is shown relative to expression of these genes in the matched drug-naïve cells. Bars represent mean values of 3 biological replicates ±SD, except for 12_PLXR cells (*n* = 2). Differences are considered significant at * *p* < 0.05. Lower part: transcript level in each cell line was related to the median value calculated for all 11 resistant cell lines. (**C**) Proteins levels of SOX2 and SOX10 were determined in drug-naïve, PLXR and TRAR cell lines by Western blotting. GAPDH was used as a loading control; *n* = 3, except for 12_PLXR cells (*n* = 2).

**Figure 3 cells-09-00142-f003:**
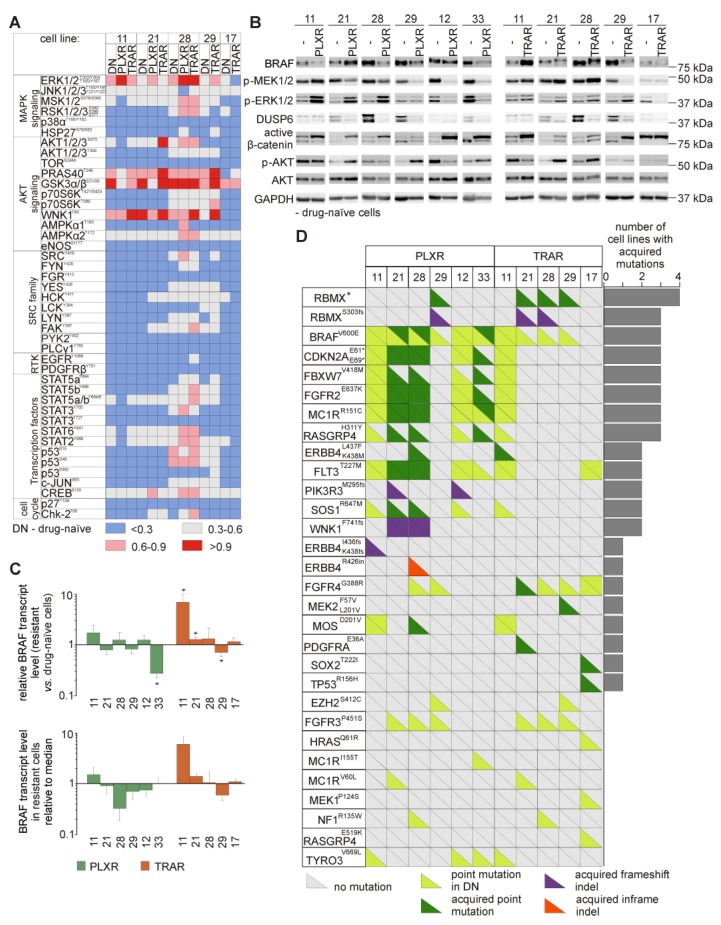
Melanoma cells resistant to vemurafenib (PLXR) or trametinib (TRAR) differ from their drug-naïve counterparts in activities of the main signaling pathways, which might be partially explained by acquired mutations. (**A**) Differential activation of key signaling molecules from drug-naïve (DN) and drug-resistant (PLXR, TRAR) melanoma cells assessed by phospho-profiling. The graph shows relative levels of phospho-proteins, for which mean pixel densities were normalized to HSP60 level. Red and blue colors indicate high- and low-level phosphorylation, respectively. (**B**) Western blots showing differences in expression of BRAF, DUSP6, and AKT and phosphorylation of MEK1/2, ERK1/2, β-catenin, and AKT. GAPDH was used as a loading control. *n* = 3, except for 12_PLXR cells (*n* = 2). (**C**) BRAF transcript levels were assessed using qRT-PCR. Upper part: expression of BRAF in PLXR and TRAR cell lines relative to its expression in the matched drug-naïve cells. Bars represent mean values of 3 biological replicates ±SD, except for 12_PLXR cells (*n* = 2). Differences are considered significant at * *p* < 0.05. Lower part: transcript levels of BRAF determined in drug-resistant cells relative to the median value. (**D**) Color coded diagram of mutations acquired during the development of resistance. Mutations in genes encoding proteins of the MAPK, PI3K/AKT, RTK, and cell cycle pathways are included. See Appendix A for non-mutated genes. All mutations (SNPs and InDELs) are ranked according to the number of resistant of cell lines, in which they were detected. RBMX *: G379R+/−, Y357H+/−, S337N+/−, R339G+/−, R324P+/−.

**Figure 4 cells-09-00142-f004:**
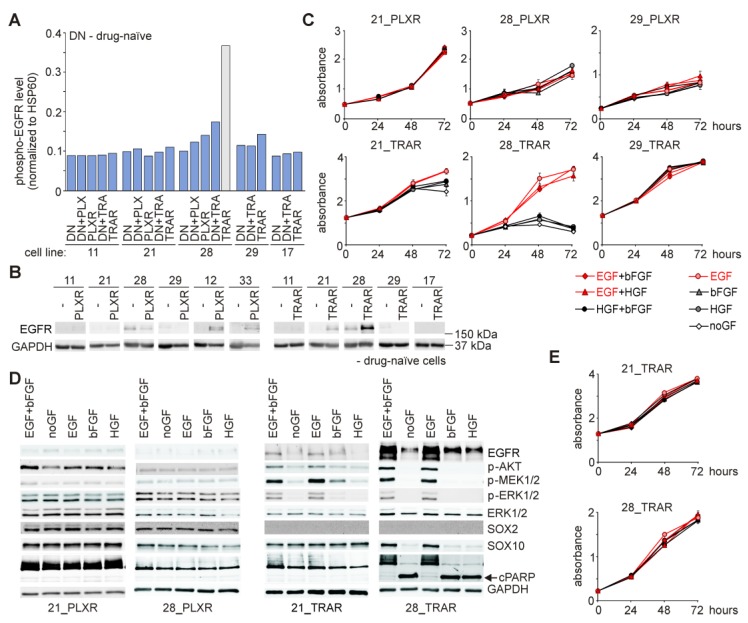
Growth factor dependence is not frequently developed by resistant melanoma cells, and EGFR-expressing TRA-resistant cells exert EGF-dependence only in the presence of drug. (**A**) Phospho-EGFR levels in drug-naïve, vemurafenib-resistant (PLXR) and trametinib-resistant (TRAR) cells were assessed by phospho-profiling and normalized to a HSP60 level. (**B**) Representative immunoblots showing *EGFR* expression in drug-naïve, PLXR and TRAR cell lines. GAPDH was used as a loading control; *n* = 3, except for 12_PLXR cells (*n* = 2). (**C**) Proliferation curves obtained based on the activity of acid phosphatase in PLXR or TRAR cells treated with the respective drug while growing in the presence of different growth factors or without any of them (noGF). (**D**) Selected resistant cells lines (21_PLXR, 28_PLXR, 21_TRAR, and 28_TRAR) were treated with the respective drug for 19 h in the presence of different growth factors or without any of them (noGF). Representative immunoblots showing total and phosphorylated protein levels are shown. GAPDH was used as a loading control; *n* = 3. (**E**) 21_TRAR and 28_TRAR cell lines were grown in trametinib-free medium in the presence of different growth factors or without any of them (noGF). Proliferation curves were obtained based on activity of acid phosphatase. Representative results of experiment performed in triplicate are shown.

**Figure 5 cells-09-00142-f005:**
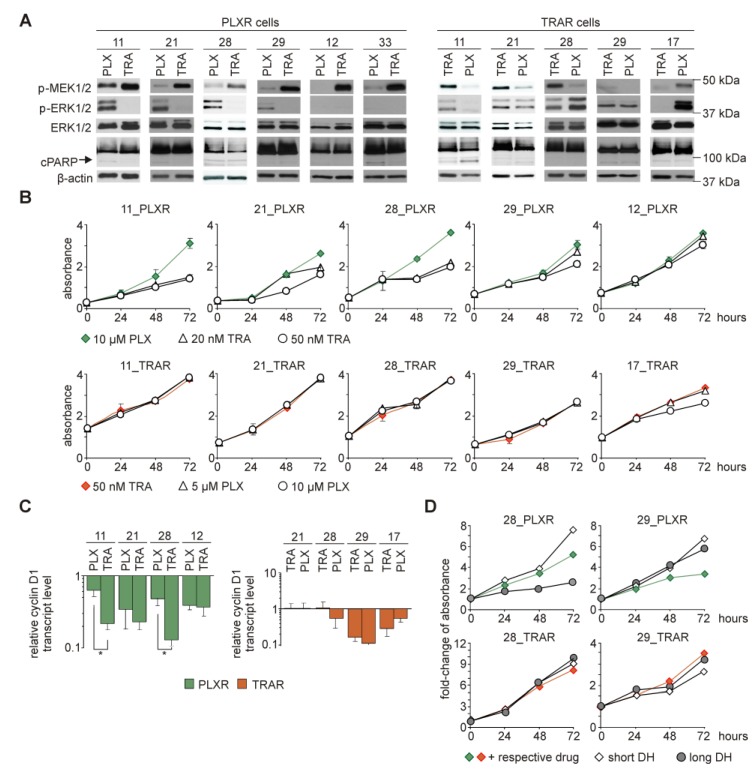
A remarkable plasticity of resistant melanoma cells shown as phenotypic reprogramming accompanying development of cross-resistance. Vemurafenib-resistant (PLXR) cells were shortly exposed to trametinib (TRA), whereas trametinib-resistant cells (TRAR) to vemurafenib (PLX) at indicated concentrations. (**A**) Representative immunoblots showing levels of phosphorylated MEK1/2 (p-MEK1/2) and ERK1/2 (p-ERK1/2), and cleaved PARP (cPARP) after 19 h of treatment with a corresponding drug. β-actin was used as a loading control. (**B**) Proliferation curves of PLXR and TRAR cell lines grown in the presence of different concentrations of complementary drugs were obtained based on activity of acid phosphatase. Representative results of an experiment performed in triplicate are shown. (**C**) Cyclin D1 transcript levels were assessed using qRT-PCR. They were expressed relative to their levels in drug-naïve cells cultured without drugs. Bars represent mean values of 3 biological replicates ±SD. Differences are considered significant at * *p* < 0.05. (**D**) PLXR and TRAR cell lines were either exposed to the respective drug (10 µM PLX or 50 nM TRA) or subjected to either short drug holiday (short DH) or drug holiday for 10 days (long DH). Fold-changes of absorbance values at different time points (24 h, 48 h, 72 h) were calculated relative to absorbance value at the starting point of the APA assay (0 h). Representative results of an experiment performed in triplicate are shown.

**Figure 6 cells-09-00142-f006:**
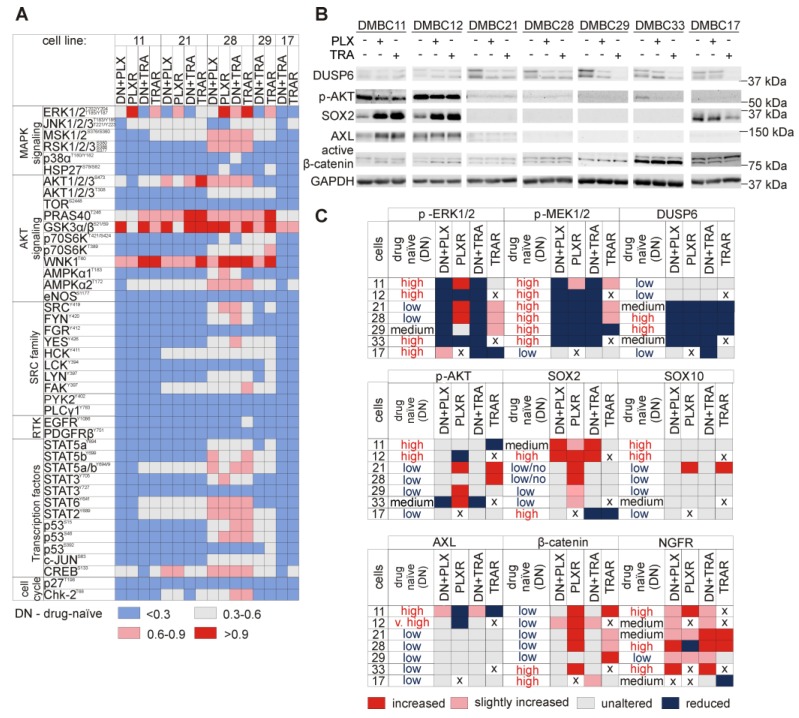
Comparison of vemurafenib (PLX)- or trametinib (TRA)-resistant cells with their drug-naïve counterparts shortly exposed to drugs revealed that several resistance-attributed alterations are already induced during short treatment. (**A**) Differential activation of key signaling molecules in drug-naïve (DN) melanoma cells after short treatment with drugs and drug-resistant (PLXR, TRAR) melanoma cells assessed by phospho-profiling. The graph shows relative levels of phospho-proteins, for which mean pixel densities were normalized to HSP60 level. Red and blue colors indicate high- and low-level of phosphorylation, respectively. (**B**) Western blot showing differences in the levels of DUSP6, SOX2, and AXL, and phosphorylation of AKT and dephosphorylated β-catenin between drug-naïve melanoma cells and cells shortly exposed to drugs. GAPDH was used as loading control; *n* = 3. (**C**) Color coded tables comparing expression and activity of proteins in melanoma cells during immediate response to drugs (DN+PLX; DN+TRA) or in stable drug-resistant cells (PLXR, TRAR) with respect to drug-naïve cells. Data are extracted from analysis of phospho-kinase array, immunoblotting, and flow cytometry presented in the figures in this report and elsewhere [22,27]. Changes in the level/activity of proteins are color coded (red, increase; blue, decrease; grey, no change).

**Figure 7 cells-09-00142-f007:**
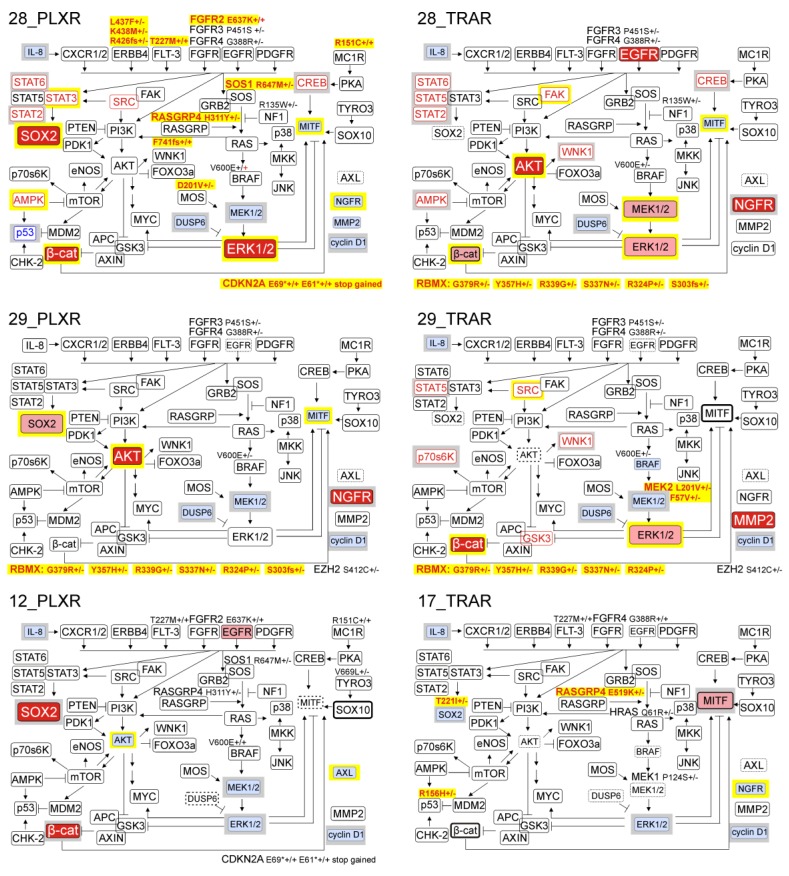
Diverse patterns of resistance showing patient- and drug-specific differences between drug-naïve and drug-resistant melanoma cells. The profiles summarizing alterations in the major signaling pathways were prepared for selected cell lines. The scheme showing changes in all resistant cells lines along with methods of data acquisition is provided as Appendix A. Color coded overview comparing investigated cell lines in terms of differences between drug-naïve and drug-resistant cells. Red/pink, strong/weak enhancement; blue, inhibition; red/blue frame and font, enhancement/inhibition shown only in phospho-profiling; dotted frame smaller in size, weak level/activity in drug-naïve cells; bold frame, high level/activity in drug-naïve cells; mutations acquired by resistant cells are marked in red, and those already present in drug-naïve cells are marked in black. Changes, which were already induced during immediate response to drugs are in the grey background, whereas changes unique for resistant cells (including mutations) are in the yellow background.

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
