# Peer review of "Dissecting Mechanisms of Melanoma Resistance to BRAF and MEK Inhibitors Revealed Genetic and Non-Genetic Patient- and Drug-Specific Alterations and Remarkable Phenotypic Plasticity"

_cells, 2020, doi:10.3390/cells9010142_

Round 1
Reviewer 1 Report
Comments:
This manuscript by Hartman et al. seeks to identify the mechanisms related to MAPK pathway inhibitors in melanoma. Using patient derived cell lines the authors generate resistant models using both the Braf inhibitor vemurafenib and the MEK inhibitor trametinib. They go on to further demonstrate the potential mechanisms of resistance to these inhibitors in these cell lines using both protein methods and whole exome sequencing. Through these methodologies they then compared the intrinsic resistance to previously described mechanisms of resistance including ALK, EGFR, MITF etc. Overall this is an intriguing study with strong efforts having been made to understand the unique mechanisms of resistance which may be utilized by melanoma tumours to overcome MAPK inhibition. Although the study is performed to a high standard and is well written, the flow of the article is extremely difficult to follow with the authors attempting to relate each resistance cell line to every single possible resistant mechanism. The article needs to be formatted in way for the readers to easily follow the data. This is understandably very difficult to do as there are no obvious trends for any of the cell lines. Besides the important point the authors are trying to make concerning AXL/MITF levels it may be suggested to focus on the changes which specially relate to resistance rather then also highlighting changes that don’t. ie. BRAF transcript levels in in 33-PLXR and 29-TRAR are reduced. This may be an important point but not necessarily related to MAPKi resistance. Other concerns are listed below.
Major concerns
The authors need to demonstrate that at least one of the potential mechanisms of resistance is relevant. Does co-treatment with a CDK inhibitor, WNT inhibitor or PI3K inhibitor enhance cell death in those cell lines with increased Cyclin D or p-AKT or active B-catenin compared to those resistant cell lines that don’t have these pathways activated.
Truly acquired new mutations is extremely rare in resistant models. Usually these rare genetic variants exist in a subpopulation of cells. The finding that PLXR 21 and 28 have acquired a number of new point mutations is important and should be discussed.
The authors discuss both genetic and non-genetic factors associated with MAPKi resistance however they did not explore if these factors are reversible. It is essential that the authors demonstrate that if vemurafenib and/or trametinib treatment is discontinued that the cell lines revert to their original drug naïve state in terms of protein expression (Fig 3B). This is a critical question as more than likely a few of these cells will die without MAPK inhibition. Therefore, potentially defining a link between these resistant mechanisms and a “drug holiday”.
Although the authors state pten deletion was not detected in any of their cell lines, a pten western needs to be performed for figure 3B. Pten expression can be regulated through multiple mechanisms.
Combination treatment with plx and tramitnib needs to be done on a few cell lines to show combination treatment is effective in killing these cell lines.
Minor concerns
Figure 1A is of poor quality. A higher quality image and possibly a larger image is suggested.
‘ Increasing numbers of viable cells over time indicate that PLXR and TRAR cells became insensitive to drugs in contrast to their original counterparts.” T29 and T 17 appear to be relatively equal in sensitivity.
The reason the authors looked at cyclin d needs to be discussed.
Figure 1C Please place the drug naïve state at the top followed by plxr and trar.
Please state that the MITF results were previously published in the main text not just reference
Figure 3a. In the figure lines are missing between the cells lines making it hard to decipher what cell is treated with what. Use bold lines to distinguish between cell lines.
Figure 4C either the figure needs to be clearer or made bigger. It is impossible to determine what line is what.
Figure 7 is an inappropriate figure for a journal. Either choose one cell line and place the rest in Supplemental figures or place all figure 7 results in supplemental figures.
Soufiane Boumahdi and Frederic J. de Sauvage. The great escape: tumour cell plasticity in resistance to targeted therapy. Nature Reviews Drug Discovery 2019 is suggested to be referenced.
Author Response
Response to comments of Reviewer 1
This manuscript by Hartman et al. seeks to identify the mechanisms related to MAPK pathway inhibitors in melanoma. Using patient derived cell lines the authors generate resistant models using both the Braf inhibitor vemurafenib and the MEK inhibitor trametinib. They go on to further demonstrate the potential mechanisms of resistance to these inhibitors in these cell lines using both protein methods and whole exome sequencing. Through these methodologies they then compared the intrinsic resistance to previously described mechanisms of resistance including ALK, EGFR, MITF etc. Overall this is an intriguing study with strong efforts having been made to understand the unique mechanisms of resistance which may be utilized by melanoma tumours to overcome MAPK inhibition. Although the study is performed to a high standard and is well written, the flow of the article is extremely difficult to follow with the authors attempting to relate each resistance cell line to every single possible resistant mechanism. The article needs to be formatted in way for the readers to easily follow the data. This is understandably very difficult to do as there are no obvious trends for any of the cell lines. Besides the important point the authors are trying to make concerning AXL/MITF levels it may be suggested to focus on the changes which specially relate to resistance rather then also highlighting changes that don’t. ie. BRAF transcript levels in in 33-PLXR and 29-TRAR are reduced. This may be an important point but not necessarily related to MAPKi resistance. Other concerns are listed below.
We appreciate this very nice summary of the main goals and achievements of our study. We also would like to thank the Reviewer for thoughtful revision of our manuscript and valuable critical comments. We think that by addressing the Reviewer’s concerns described below, we have substantially improved the clarity and impact of our manuscript.
We agree with the Reviewer that there are no obvious trends for any of the cell lines acquiring drug resistance. That is the point of our study, and the flow of our manuscript can be considered difficult but we had to provide relatively large amount of data to evidence variability in expression and activity of a number of molecules contributing to resistant phenotype of melanoma cells, as well as remarkable plasticity of resistant cells. This was necessary to support the conclusion that patient-to-patient variability is richer and more nuanced than previously described. However, we also agree with the Reviewer that there were several sentences describing results unrelated to observed resistance, and therefore several of them were removed. To enhance the significance of our study in a convincing manner and to enable the readers to more easily understand general conclusions provided by this report, we summarized the main findings in the Figure 7, which has been now modified according to the Reviewer’s suggestion. This figure but especially Figure S2 can be considered difficult to be deciphered, but they reflect very well the overall conclusion i.e. substantial patient-to-patient and drug-to-drug diversity of genetic and non-genetic alterations accompanying resistance to targeted therapies. We have attempted to focus on most relevant aspects of melanoma cell phenotype in a light of current literature. We have also addressed all Reviewer’s concerns (see below), however, suggestions to provide additional results to this already very reach in data manuscript have been treated as very interesting hints for the future studies, and we fulfilled the expectations of the Reviewer’s in this respect only partially.
Major concerns:
The authors need to demonstrate that at least one of the potential mechanisms of resistance is relevant. Does co-treatment with a CDK inhibitor, WNT inhibitor or PI3K inhibitor enhance cell death in those cell lines with increased Cyclin D or p-AKT or active B-catenin compared to those resistant cell lines that don’t have these pathways activated.
Actually, we validated one of the potential mechanisms of resistance by performing a functional assay to demonstrate how deprivation of defined growth factor, i.e. EGF (epidermal growth factor) can affect survival of resistant melanoma cells that acquired the ability to express EGFR at a high level. It was intriguing because we have previously reported that EGF, but also bFGF and HGF, did not affect the response of drug-naïve melanoma cells to short exposure to vemurafenib or trametinib (Zalesna et al., 2017 PloS ONE). We think that deprivation of a selective ligand for the receptor, which does not harbor any genetic alteration that could possibly drives ligand-independent activity, can be considered as an effect of inhibition of receptor activity. Concerning a robust cell death assessed in EGFRhigh 28_TRAR cells (Figure 4D), these experiments validate one of potential mechanisms of resistance, and can be further exploited e.g., in studies assessing the effectiveness of combinatorial MAPK and EGFR inhibition in drug-resistant cell lines, in which the EGFR addiction was acquired.
Truly acquired new mutations is extremely rare in resistant models. Usually these rare genetic variants exist in a subpopulation of cells. The finding that PLXR 21 and 28 have acquired a number of new point mutations is important and should be discussed.
It cannot be excluded that mutations found exclusively in resistant cell lines are a result of the outgrowth of resistant subclones constituting a very minor subpopulation already present in matched drug-naïve cell lines according to Darwinian selection model. As the raw data of WES are available at ArrayExpress and the European Nucleotide Archive (ENA), they can be further explored, and the genetic variants found in our study can be assessed at the level of single cells and/or tracked during the development of resistance. We agree with Reviewer that this observation is important, and we think that the summary provided as Figure 7 and Fig. S2 clearly supports the conclusion that 21_PLXR and 28_PLXR cell lines acquire a substantial number of genetic variants compared with other resistant cell lines.
The authors discuss both genetic and non-genetic factors associated with MAPKi resistance however they did not explore if these factors are reversible. It is essential that the authors demonstrate that if vemurafenib and/or trametinib treatment is discontinued that the cell lines revert to their original drug naïve state in terms of protein expression (Fig 3B). This is a critical question as more than likely a few of these cells will die without MAPK inhibition. Therefore, potentially defining a link between these resistant mechanisms and a “drug holiday”.
We have included additional results showing how proliferation of resistant melanoma cell lines is affected by drug holiday. A drug discontinuation for 19 h did not induce cell death (Figure 5A) and drug withdrawal for up to 3 days (short drug holiday) did not affect the viable cell number in any of four representative cell lines (Figure 5D in the revised version of manuscript). The shape of the proliferation curves was very similar for drug-treated and untreated (short drug holiday) resistant cells (Figure 5D), and cells after drug withdrawal did not proliferate slower and even a higher proliferation rate was observed in case of 28_PLXR cells. These new results could be used to support the conclusion that we observed the inhibitory effect of trametinib in 28_PLXR cells in the cross-resistant experiment, not vemurafenib withdrawal effect (Figure 5B). They also support the statement that observed changes in the activities of MEK1/2 and ERK1/2 are the results of cell reprograming, not selection. Moreover, proliferation of 29_PLXR, 28_TRAR and 29_TRAR cells remained unaffected when these resistant cells subjected to drug holiday for 10 days (long drug holiday) were used in the assay (Figure 5D). Although proliferation of 28_PLXR cells on long drug holiday was slower, a time-dependent increase in viable cell number was still observed (Figure 5D). By including results of these experiments, we have addressed the critical question given by the Reviewer, and we have shown that resistant cell lines do not die, when MAPK inhibition was suspended. In addition, in our recently published paper (Czyz et al., 2019 J Oncol – Fig. 7a and Supp. Fig. S1), we have demonstrated lack of PARP cleavage in resistant cells subjected to long drug holiday. We have also shown that long drug holiday affected MITF level and expression of MITF-dependent pigmentation-related genes (Fig. 7a & b in Czyz et al., 2019). After drug withdrawal, however, their expression reached the levels observed in drug-naïve cells only in 29_TRAR and 17_TRAR cells, whereas in other resistant cells it was not substantially changed. As we expect a lot of variations for other signaling pathways, a comprehensive study of alterations that might accompany “drug holiday” cannot be in scope of our present study.
Although the authors state pten deletion was not detected in any of their cell lines, a pten western needs to be performed for figure 3B. Pten expression can be regulated through multiple mechanisms.
We agree that PTEN can be modulated at different levels as multiple mechanisms have been shown to affect PTEN expression, including gene mutations and deletions, regulations at transcriptional, epigenetic, posttranscriptional, posttranslational and protein-protein interaction levels (Bermudez et al., 2015 Front Oncol; Liu et al., 2019 Cancers). PTEN is recognized as a tumor suppressor that antagonizes PI3K/AKT signaling through its phosphatase activity (Li et al., 1998 Cancer Res). In addition, PTEN can function independently of its phosphatase activity, and it plays a role in the nucleus and can be secreted to extracellular matrix (Bermudez et al., 2015 Front Oncol; Liu et al., 2019 Cancers).
PTEN deletion, which has been assigned to almost 44% of BRAF-mutated and 4% of NRAS-mutated melanomas (Hodis et al., 2012 Cell), was not present in the genome of any investigated melanoma cell line, either drug-naïve or drug-resistant. In a more recent study (Giles et al., 2019 J Invest Dermatol), only 18 of 287 tumors (6.3%) had PTEN deletions, and 24 of 287 tumors (8.4%) carried PTEN mutations suggesting that genetic alterations in PTEN can be relatively uncommon in melanoma. Giles et al. also concluded that “PTEN genomic alterations (deletion, mutation), promoter methylation, and protein destabilization did not fully explain PTEN loss in melanoma” (Giles et al., 2019 J Invest Dermatol), which supports the complexity of PTEN regulation in melanoma. In addition, the phenotypic outcome of a decrease in PTEN level can be differentially manifested depending on the extent of PTEN downregulation, as well as cell type and genetic background eg., TP53 alterations (Berger et al., 2011 Nature). In our study, we focused on assessing the level of phospho-AKT, which is an effector kinase that acts downstream of PTEN. We mentioned lack of PTEN deletion to only exclude this mechanism of AKT activation in phospho-AKThigh drug-resistant cell lines, however, as PTEN deletion was not the case in any of resistant cell lines, we removed this information following the Reviewer’s suggestion not to discuss mechanisms obviously unrelated to resistance in our settings.
Combination treatment with plx and tramitnib needs to be done on a few cell lines to show combination treatment is effective in killing these cell lines.
This experimental setting suggested by the Reviewer was included in our study, especially regarding newly provided data (Fig. 5D in the revised version of manuscript). We have grown trametinib-resistant (TRAR) cells in the presence of vemurafenib (and vice versa), which means that TRAR cells with established resistant phenotype were exposed to vemurafenib (and vice versa). For four representative cell lines (Fig. 5D), and in our recent report (Czyz et al., 2019 J Oncol – Fig. 7A and Supp. Fig. S1), we have demonstrated that drug cessation does not affect viability of drug-resistant cell lines as confirmed by lack of PARP cleavage and unaffected proliferation curves. This indicates that changes assessed at the molecular (Fig. 5A and C) and cellular (Fig. 5B) levels during cross-resistance experiments resulted exclusively from response of resistant cells to a complementary drug. Owing to this approach, we provided a few interesting observations concerning molecular alterations within the activity of the MAPK signaling pathway and their consequences on melanoma cell proliferation and phenotype. Especially interesting is the observation that while ERK1/2 activity is suppressed by trametinib in vemurafenib-resistant cells that exerted the high activity of these kinases, activity of MEK1/2 is enhanced. In addition, it is clearly visible that elevated level of p-ERK1/2 in vemurafenib-resistant melanoma cells can be reversed, without inducing cell death (no cPARP) but with inhibiting cyclin D1 expression and proliferation. This was not observed in trametinib-resistant cell lines exposed to vemurafenib.
We also enhanced the quality of graphs in the panel B of Figure 5.
Minor concerns:
Figure 1A is of poor quality. A higher quality image and possibly a larger image is suggested.
According to the Reviewer’s suggestion, we modified the graphs and enhanced the quality of this panel.
‘Increasing numbers of viable cells over time indicate that PLXR and TRAR cells became insensitive to drugs in contrast to their original counterparts.” T29 and T17 appear to be relatively equal in sensitivity.
We understand that drug-resistant means that cells are capable to grow in the presence of drug at the concentration, which is cytotoxic to drug-naïve cells. In our study, the resistant cells are not drug-tolerant cells, which were selected after short exposure to drugs. Resistant cells can proliferate continuously in the presence of drug at high concentration. It is well defined in the recent review article of Rambow et al., on melanoma phenotypic plasticity (2019, Genes Dev): “as opposed to drug resistance, drug-tolerance relates to a state in which tumor cells can survive, but not proliferate during treatment”. Although the difference in proliferation curves in 29_TRAR and 17_TRAR cell lines relative to drug-naïve cells grown in the presence of drug is less marked, it is visible. In DMBC29 (drug-naïve) cells, trametinib inhibits proliferation after 48-72 hours as no further increase in viable cell number is observed over time in contrast to 29_TRAR cells. In case of 17_TRAR and DMBC17 cells grown in the presence of trametinib, the difference is visible after 72 hours. Interestingly, these two cell lines are MITFhigh and express several differentiation/pigmentation-related markers at a high level, both as drug-naïve and trametinib-resistant cell lines in contrast to other paired drug-naïve/drug-resistant cell lines [Figure 5 and Figure 6 in Czyz et al., 2019].
Based on the Reviewer’s suggestion, we changed the description of the results accordingly.
The reason the authors looked at cyclin d needs to be discussed.
Cyclin D1, whose expression is induced by the Ras/Raf/ERK pathway, has been identified as a part of cell cycle-regulating signaling pathway involved in melanomagenesis (Bartkova et al., 1996 Cancer Res; Maelandsmo et al., 1996 Br J Cancer; Young et al., 2014 PCMR). Upregulation of cyclin D1 as an effect of genomic amplification has been also demonstrated as one of the mechanisms of resistance to BRAF inhibitor (Smalley et al., 2008 Mol Cancer Ther). In addition, we have reported that cytostatic effect of vemurafenib and trametinib on drug-naïve cells is uniformly accompanied by significant decrease in the level of cyclin D1 mRNA (Hartman et al., 2017 Lab Invest). Therefore, it was justified to assess expression of cyclin D1 as one of the putative mechanisms of resistance in melanoma cells. An extended explanation is included in the revised version of the manuscript.
Figure 1C Please place the drug naïve state at the top followed by plxr and trar.
Figure 1C was modified accordingly.
Please state that the MITF results were previously published in the main text not just reference
It was modified in the main text, and the statement that “The original data for MITF was published elsewhere” is included in the legend of the Figure 1.
Figure 3a. In the figure lines are missing between the cells lines making it hard to decipher what cell is treated with what. Use bold lines to distinguish between cell lines.
Figure 3A was modified.
Figure 4C either the figure needs to be clearer or made bigger. It is impossible to determine what line is what.
We enhanced the quality of this panel, and 4E in addition. To make the graphs clearer, we also colored the lines corresponding to EGF-containing conditions.
Figure 7 is an inappropriate figure for a journal. Either choose one cell line and place the rest in Supplemental figures or place all figure 7 results in supplemental figures.
This summary Figure 7 is not to depict particular information in details as all included data can be found in other figures or supplementary material. Overall message focusing on the patient-to-patient and drug-to-drug variability of resistance–associated alterations was obtained by using different fonts and colors. We think that while it is not a typical (or standard) way of presenting data, it is the right way to show a high variability of resistant phenotypes. However, following the Reviewer’s suggestion, we moved the original Figure 7 to the Supplementary Material (Fig. S2 in the revised version of the manuscript). In addition, we have chosen six representative drug-resistant cell lines that well reflect the variability of genetic and non-genetic alterations accompanying acquired resistance. In this way, the selected schemes could be substantially enlarged, and some additional modifications were introduced to enhance the clarity of this figure.
Soufiane Boumahdi and Frederic J. de Sauvage. The great escape: tumour cell plasticity in resistance to targeted therapy. Nature Reviews Drug Discovery 2019 is suggested to be referenced.
We would like to thank the Reviewer for calling our attention to this recently released paper, which we obviously overlooked. While it is about general tumour cell plasticity in resistance to targeted therapy, it provides very interesting information on melanoma plasticity as well. Therefore, this review was referenced and highlighted in the revised version of our manuscript.
Reviewer 2 Report
In this manuscript Hartman and colleagues have established a model of melanoma resistance to vemurafenib or trametinib to provide insights into mechanisms leading to acquired resistance. Authors have addressed an important question related to the mechanisms leading to development of acquired resistance to BRAF and MEK inhibitors in melanoma. The manuscript is technically sound and the quality of presentation is good. However, the manuscript looks like a collection of results, and it does not add much to what already published on the subject. It would benefit a lot the manuscript if authors could investigate more deeply some of the findings. For instance, it would be interesting to investigate the biological relevance of the frameshift variant of RBMX found exclusively in phospho-AKT high resistant cell lines. Does this frameshift variant of RBMX contribute to the increased activity of AKT in melanoma cells?
In Figure 2C authors show that SOX2 expression is consistently upregulated in PLXR resistant cell lines compared to drug naïve. This finding is at odd with results shown by Cesarini et al. (Oncogene. 2017 Aug;36(31):4508-4515. doi: 10.1038/onc.2017.53). Authors should provide an explanation for this discrepancy.
A significant concern related to this manuscript is that there are not melanoma cell lines (such as A375) to which one can relate to. The fact that there are no similar pattern of phenotypic alterations among eleven resistant cell lines, including expression/activity of crucial regulators, such as MITF, AXL and NGFR is very odd.
Other points:
Figure 1A should be enlarged because it is not readable. In Figure 3A and 6A how were cut-off chosen (<0.3, 0.3-0.6, 0.6-0.9, >0.9). Human Phospho-Kinase Array membranes should be shown in Supplementary Information (Related to Figures 3A and 6A).Author Response
Response to comments of Reviewer 2
In this manuscript Hartman and colleagues have established a model of melanoma resistance to vemurafenib or trametinib to provide insights into mechanisms leading to acquired resistance. Authors have addressed an important question related to the mechanisms leading to development of acquired resistance to BRAF and MEK inhibitors in melanoma. The manuscript is technically sound and the quality of presentation is good. However, the manuscript looks like a collection of results, and it does not add much to what already published on the subject. It would benefit a lot the manuscript if authors could investigate more deeply some of the findings. For instance, it would be interesting to investigate the biological relevance of the frameshift variant of RBMX found exclusively in phospho-AKT high resistant cell lines. Does this frameshift variant of RBMX contribute to the increased activity of AKT in melanoma cells?
We appreciate the Reviewer’s positive comments and critical judgement. It is retrospectively evident that some of our statements were not clearly presented. Accordingly, we have comprehensively addressed all the criticisms and revised the manuscript. As scientists with different background, we are evaluating and judging results from different perspectives. We would like to convince the Reviewer that our standpoint is credible based on the relevant literature and our own results.
An important message that can be derived from our study is a high variability of molecular and cellular changes induced during development of drug resistance. There was no similar pattern of genetic/non-genetic alterations among eleven investigated resistant cell lines, including phenotypic changes in expression/activity of crucial melanoma regulators e.g., MITF, AXL, NGFR, AKT, ERK1/2 and SOX2. Vemurafenib- and trametinib-resistant cells differed even if they were obtained from the same patient-derived drug-naïve melanoma cell line. This means that most of genetic and non-genetic alterations are triggered in a cell line- and/or drug-specific manner. Several changes previously assigned to the development of resistance were induced already by short treatment to the extent measurable at the bulk levels, which means that these alterations observed during initial response do not affect only a small subpopulation of melanoma cells. Moreover, the results presented in this report indicate that melanoma cells, drug-naïve but also drug-resistant present remarkable plasticity. EGF-dependence was limited to the resistant cell line with elevated level of EGFR expression/phosphorylation, and it was conditional as excluding trametinib from the culture medium lacking EGF resulted in the rapid regrowth and reprograming of melanoma cells. Alterations observed in the cross-resistance experiments, best exemplified by phosphorylation of MEK1/2 in vemurafenib-resistant cells after short treatment with trametinib also indicate a high plasticity of resistant melanoma cells. In addition to assessing the level/activity of a plethora of molecules to broadly capture the phenotype of melanoma cells, we have also performed a functional assay to validate dependence of trametinib-resistant cell lines on EGF.
Our study provides also specific conclusions supplementing the current knowledge. (1) AXL is not an unambiguous marker of resistance to targeted therapeutics developed in melanomas. (2) Increased percentages of NGFR (CD271)-positive cells and reduced percentages of MITF-positive cells is true for the majority but not all resistant melanoma cell populations. Therefore, the results showing a positive or negative alignment of expression or activity of two or more proteins should be interpreted cautiously as they might apply only to selected patients. (3) For the first time, we have reported a frameshift variant of RBMX in melanoma. This alteration was present exclusively in resistant cell lines with an elevated activity of AKT, which should be further investigated as a potential mutation-based route of melanoma resistance to targeted therapeutics. It is not in scope of our study, which is mainly about contribution of phenotypic plasticity to drug resistance, to validate the role of variants of RBMX but by sharing this observation with scientific community, we could stimulate someone to undertake the mechanistic study.
In Figure 2C authors show that SOX2 expression is consistently upregulated in PLXR resistant cell lines compared to drug naïve. This finding is at odd with results shown by Cesarini et al. (Oncogene. 2017 Aug;36(31):4508-4515. doi: 10.1038/onc.2017.53). Authors should provide an explanation for this discrepancy.
In a study by Cesarini et al. (2017, Oncogene), only three vemurafenib-resistant cell lines were investigated and substantial upregulation of SOX2 expression was detected in one out of these three resistant melanoma cell lines (Fig. 4f therein). In another study, SOX2 upregulation was also found in one out of three melanoma cell lines with acquired resistance to vemurafenib (Supplementary Fig. 9 in Huser et al., 2018 Int J Cancer). Both studies indicate that an increase of SOX2 in vemurafenib-resistant cells is possible, however, different frequency of this phenomenon was found than in our study. It is worth to note that there is a discrepancy in the levels of SOX2 in A375 cells between these two reports: SOX2 is not expressed in both drug-naïve and drug-resistant A375 cells by Cesarini et al., (Oncogene,2017), while SOX2 is clearly detectable in A375 cells in a study by Huser et al (Fig. 4 and Supplementary Fig. 9 in Huser et al., 2018 Int J Cancer, 2018).
In a study by Cesarini et al. (2017, Oncogene), it was demonstrated that “both SOX2-positive and SOX2-negative cells reached vemurafenib or dabrafenib resistance independently from SOX2 expression“. Huser et al reported a marked increase of SOX2 level after acute exposition to vemurafenib, and SOX2 regulated expression of CD24 to drive adaptive resistance to vemurafenib (Fig. 1e in Huser et al., 2018 Int J Cancer). In addition, increased level of SOX2 was assessed in undifferentiated melanoma cells that were resistant to inhibitors of the MAPK signaling pathway (Bernhardt et al., 2017 Stem Cell Rep). Thus, SOX2 may contribute to maintenance of the resistant phenotype of PLXR cells (this study) and additionally adapt melanoma cells to acute exposure to drug in certain cell lines. This is supported by the results showing that an increase of SOX2 level in response to a short exposure of drug-naïve cells to vemurafenib was found in DMBC11 and DMBC12 cell lines (Fig. 6B in our study). Nevertheless, the resistant cell lines (all PLXR) obtained as the result of growing melanoma cells for 4-5 months in the presence of vemurafenib demonstrated an enhanced level of SOX2 (Fig. 2B and 2C). This was clearly visible at the transcript (Fig. 2B) and protein (Fig. 2C) levels. According the Reviewer’s suggestion, discussion of results of Cesarini et al on SOX2 was extended in the revised version.
A significant concern related to this manuscript is that there are not melanoma cell lines (such as A375) to which one can relate to. The fact that there are no similar pattern of phenotypic alterations among eleven resistant cell lines, including expression/activity of crucial regulators, such as MITF, AXL and NGFR is very odd.
Although A375 cell line is broadly used in the laboratories, it has been established a long time ago and many variants of this cell line exist giving contradictory results (one example is mentioned above in response to the Reviewer’s comment). This cell line is MITFlow, and could potentially serve as the reference to DMBC11 and DMBC12 cell lines. But why should we do this? The major problem in finding the right treatment for melanoma patients is high heterogeneity and plasticity of this tumor. Therefore, any action ignoring diversity observed in melanoma is not the right one. We have obtained 11 melanoma cells lines, which is a lot if you compare with other studies. And each cell line is different, however, if we wish very much to get more uniform picture, we could have probably done this by excluding some cell lines and selected results, which are in line with the chosen hypothesis. But is it the right way? We have to accept that this is not odd that so many variants of melanoma exist and the pathways to reach drug resistance are also very diverse. And our work is about diversity and plasticity of patient-derived melanoma cell lines and their drug-resistant counterparts. In our study, we took advantage of having patient-derived drug-naïve cell lines that were widely characterized at the molecular and cellular levels (Sztiller-Sikorska et al., 2015 Lab Invest; Hartman et al., 2016 Oncotarget; Hartman et al., 2017 Lab Invest; Zalesna et al., 2017 PloS ONE; Mielczarek-Lewandowska et al., 2019 Apoptosis; Osrodek et al., 2019 Int J Mol Sci), including data from DNA sequencing (Hartman et al., 2019 Mol Carcinog), and exhibited large heterogeneity already as drug-naïve cell lines. Our results and conclusions that “There was no similar pattern of phenotypic alterations among eleven resistant cell lines, including expression/activity of crucial regulators, such as MITF, AXL and NGFR, which suggests that patient-to-patient variability is richer and more nuanced than previously described.” are in line with the current view on the plastic nature of melanoma cell phenotype. As concluded in a recent review mentioned by the Reviewer # 1 on tumor cell plasticity and drug resistance by Boumahdi and de Sauvage (2019, Nat Rev Drug Discov) “the mechanisms controlling plasticity may vary among patients or depending on the indication or on the type and duration of the initial treatment.”
Other points:
Figure 1A should be enlarged because it is not readable.
We enlarged the graphs and enhanced the quality of this panel.
In Figure 3A and 6A how were cut-off chosen (<0.3, 0.3-0.6, 0.6-0.9, >0.9). Human Phospho-Kinase Array membranes should be shown in Supplementary Information (Related to Figures 3A and 6A).
The cut-off was set to more accurately show the differences between melanoma cell lines in respect of differential activation of key signaling molecules. Although it was an arbitrary decision, we wish to normalize the results and present unbiased comparison between samples ‘Mean pixel densities of the pair of duplicate spots representing each protein of interest and HSP60 protein (a loading control) were calculated. Relative level of each protein was calculated by using the formula:(mean pixel density of the protein)/(mean pixel density of HSP60)’, as stated in the Materials and Methods. It is notable that this way of showing data provides additional information that would be unavailable if the “n-fold change” (e.g., normalized protein level in drug-resistant versus drug-naïve cells) is calculated. Specifically, we have chosen a presentation of the results that is more absolute and enables to directly compare the extent of increase/decrease in the protein activity between different cell lines. Below we present the results of such a calculation for AKT1/2/3 phosphorylation either at S473 or at T308, which were then used for preparing graphs shown in Figure 3A and Figure 6A. Based on the given example, phospho-AKT1/2/3S473 level is 2.4-fold lower in DMBC29 (DN) cells and 4.1-fold lower in DMBC17 (DN) cells than in DMBC11 (DN) cells, suggesting diversity of baseline activity of these kinases, which can be also indirectly concluded from Western blots shown in Fig. 3B. In this case, comparing n-fold changes (normalized protein level in drug-resistant versus drug-naïve cells) would not provide information about differences between the basal levels of phosphorylated proteins observed in different melanoma cell lines.
It was technically not possible to upload this Figure to the response.
In addition, as requested by the Reviewer, we included raw images corresponding to Figures 3A and 6A in the Supplementary Material.

Round 2
Reviewer 1 Report
The authors have adequately addressed my concerns. This manuscript can now be considered suitable for publication.
Reviewer 2 Report
Overall this is an interesting study in which authors have made big efforts in order to understand and define unique mechanisms of resistance to MAPK pathway inhibition. As highlighted by the authors, it turned out that there are no obvious trends for any of the melanoma cell lines acquiring drug resistance. A high variability of molecular and cellular changes were observed, with no similar pattern of genetic/non genetic alterations among eleven investigated resistant cell lines, including phenotypic changes in crucial melanoma regulators, such as MITF, AXL, NGFR, AKT, ERK1/2 and SOX2.
I am satisfied with the changes and replies that have been made in response to my comments. The manuscript is now suitable for publication.